SciPost Physics

# Hamiltonian Truncation Crafted for UV-divergent QFTs

Olivier Delouche[1,2], Joan Elias Miró[1] and James Ingoldby[1,3]

**1** The Abdus Salam ICTP, Strada Costiera 11, 34151, Trieste, Italy
**2** Département de Physique Théorique, Université de Genève,
24 quai Ernest-Ansermet, 1211 Genève 4, Switzerland
**3** Institute for Particle Physics Phenomenology, Durham University,
Durham DH1 3LE, United Kingdom

## Abstract

We develop the theory of Hamiltonian Truncation (HT) to systematically study RG flows that require the renormalization of coupling constants. This is a necessary step towards making HT a fully general method for QFT calculations. We apply this theory to a number of QFTs defined as relevant deformations of $d = 1 + 1$ CFTs. We investigated three examples of increasing complexity: the deformed Ising, Tricritical-Ising, and non-unitary minimal model $M(3,7)$. The first two examples provide a crosscheck of our methodologies against well established characteristics of these theories. The $M(3,7)$ CFT deformed by its $\mathbb{Z}_2$-even operators shows an intricate phase diagram that we clarify. At a boundary of this phase diagram we show that this theory flows, in the IR, to the $M(3,5)$ CFT.

| | |
|---|---|
| Preprint Number | IPPP/23/79 |
| Copyright O. Delouche *et al.* | Received 15-12-2023 |
| This work is licensed under the Creative Commons | Accepted 26-03-2024 |
| Attribution 4.0 International License. | Published 19-04-2024 |
| Published by the SciPost Foundation. | doi:10.21468/SciPostPhys.16.4.105 |

# 1 Introduction

Strongly coupled Quantum Field Theories (QFTs) describe a wide array of phenomena in both condensed matter and high energy physics, but our understanding of these theories is limited by the difficulty of doing explicit calculations using them. Lattice Monte Carlo is a powerful and mature numerical approach. However not all QFTs have a lattice formulation, and for those that do, extra difficulties are encountered in particular categories of problems – such as calculations requiring real time evolution of a QFT state – which reduce the method's efficiency. As a result, the investigation of complementary numerical approaches is worthwhile.

Hamiltonian Truncation (HT) is one such promising alternative, which generalizes the Rayleigh-Ritz method from quantum mechanics. For recent reviews of its applications to QFT, see Refs. [1, 2]. To use HT, the Hamiltonian is first decomposed into a solvable part $H_0$ plus an interacting part $V$, which needn't be a weak deformation to $H_0$. Then, the eigenstates of $H_0$ with energy eigenvalues lower than a chosen HT cutoff $E_T$ are selected as a truncated basis of finite dimensionality, and the matrix elements of $V$ are found in this basis. Finally, the Hamiltonian is diagonalized numerically in this basis to yield a nonperturbative estimate for the low energy spectrum of the quantum theory. For a UV finite theory this estimate converges as the cutoff $E_T$ is raised to infinity. However for a finite value of $E_T$, the spectrum may differ significantly from the true spectrum of the theory.

Considerable progress has been made in reducing the cutoff dependence of HT calculations through the use of Effective Hamiltonians [3–13]. In these approaches, extra terms are added to the Hamiltonian that act only on low energy states. The extra terms in the Effective Hamiltonian then account for interactions with states above the cutoff that had been neglected, in an analogous way to higher dimension operators accounting for new states above the cutoff in an Effective Field Theory (EFT) Lagrangian.

A significant conceptual obstacle has been the understanding of how Hamiltonian Trunca-

tion should be applied in the case of QFTs with UV divergences that require renormalization. Superficially, the HT cutoff acts as a regulator which removes the UV divergences. Nevertheless, removing states with total energy greater than $E_T$ is a *nonlocal* modification of the theory; the maximum energy of virtual particle-antiparticle pairs that can be created at short distances becomes dependent on the energy carried by distant spectator particles. For this reason, it was demonstrated in [14] that there were extra divergences in an HT regulated QFT arising in the limit $E_T \to \infty$ that would never appear if the QFT was regulated with a local short distance cutoff regulator. Absorbing these divergences and performing renormalization requires introducing nonlocal counterterms. Finding such non-local counterterms using solely the $E_T$ regulator is challenging. There are of course infinitely many possible non-local counterterms we could consider, and we need a precise prescription for finding which are required to recover the spectrum of a local QFT as $E_T \to \infty$.

In Ref. [12], a general procedure was outlined for determining the required nonlocal counterterms in QFTs with UV divergences, which made use of an Effective Hamiltonian. In this procedure, the QFT was first regulated using a local short distance cutoff regulator and renormalized using local counterterms. An Effective Hamiltonian was then constructed for the renormalized QFT whose low energy spectrum matched that of the QFT. Finally, the local regulator was removed, which could be done analytically at the level of individual matrix elements of the Effective Hamiltonian. Extra terms left behind in the Effective Hamiltonian then provided the nonlocal counterterms needed to guarantee finiteness of the spectrum in the $E_T \to \infty$ limit.

In this paper, we revisit the procedure of [12] and apply it in numerical studies of three strongly-coupled two-dimensional QFTs as examples. Our goals are firstly to provide a demonstration of the procedure, secondly to investigate its performance and efficiency against well-studied benchmark QFTs, and then to boldly go and examine the phase structure of a new QFT, that has not been studied using HT before.

The example QFTs we consider are constructed by taking conformal field theories (CFTs) on the cylinder spacetime $\mathbb{R} \times S_{d-1}$ and deforming them using local relevant operators. The application of HT to QFTs of this kind is often referred to as the Truncated Conformal Space Approach (TCSA) [4,15,16]. In this study, we only consider deformations away from $d = 1+1$ minimal model CFTs. For these simple theories, all of the CFT data that is required to set up the HT calculation is completely known. Different minimal models $\mathcal{M}$ are labeled by two coprime positive integers $\mathcal{M}(p,q)$. For a pedagogical introduction to the minimal models, see Ref. [17].

We begin in section 2 by introducing our notation and reviewing both the TCSA and the procedure from [12]. In section 3 we present the three QFTs we will study, and discuss the TCSA in the context of minimal models. Later, in section 4 we apply our formalism to the QFT defined as the deformation of the 2d Ising CFT by its relevant $\epsilon$ operator. This QFT is related to the free massive fermion in 2d, and allows us to investigate the canceling of UV divergences and rate of convergence with $E_T$ within our approach and compare with earlier HT works. [1] Then in section 5, we apply the same analysis to the Tricritical Ising CFT deformed by its $\epsilon'$ operator. This particular deformation generates a well known integrable RG flow which ends at the 2d Ising fixed point. In section 6, we consider the $\mathcal{M}(3,7)$ CFT deformed by two of its relevant operators. This QFT is one of the simplest to feature renormalization of the coupling constant for a nontrivial interaction in its Hamiltonian. Finally, we summarize our results in section 7.

---

[1]Unlike the free fermion, the Ising does not have half integer spin operators. A subsector of bosonic states are the same for both theories, a feature that is often used for calculations. See Ref. [18] for a detailed discussion.

## 2  Hamiltonian Truncation and Effective Hamiltonians

In the rest of this work, we will determine the spectrum of QFT Hamiltonians that are defined as relevant deformations away from an ultraviolet CFT. In order to regulate infrared divergences the CFT is placed on the "cylinder" $\mathbb{R} \times S_R^{d-1}$ where $R$ is the radius of the $d-1$ dimensional sphere. Infinite volume (IR safe) observables can be obtained by taking the limit of large $R$. The Hamiltonian is given by

$$H = H_{\text{CFT}} + V \quad \text{where} \quad V = \frac{g}{S_{d-1}} \int_{S_R^{d-1}} d^{d-1}x \, \phi_\Delta(0, \vec{x}), \tag{1}$$

where $g$ is a dimensionful coupling and $\phi_\Delta$ is a relevant operator of dimension $\Delta < d$. [2] Then matrix elements of $H$ are readily computed from the knowledge of the CFT data:

$$\langle O_j | H_{\text{CFT}} | O_i \rangle = \delta_{ij} \, \Delta_i / R \qquad \text{and} \qquad \langle O_i | V | O_j \rangle = g R^{d-\Delta-1} \, \widetilde{C}_{ij}^\phi, \tag{2}$$

where $S_{d-1} = 2\pi^{d/2}/\Gamma(d/2)$ and $\Delta_i$ is the scaling dimension of operator $O_i$. We use the $\Delta_i$ notation with reference to either conformal primary, or descendant operators, or both depending on the context. In this section, we assume our basis is orthonormal for simplicity so that $\langle O_i | O_j \rangle = \delta_{ij}$. In determining the matrix elements of $V$ we used the *state→operator* map and $\widetilde{C}_{ij}^\phi \equiv \lim_{|x|\to\infty} \langle |x|^{2\Delta_i} O_i(x) \, \phi_\Delta(1) O_i(0) \rangle$, where the perturbation is evaluated at 1, which denotes a unit vector in $\mathbb{R}^d$. Our numerical work below will focus on $d = 2$ spacetime dimensions, and in deformations $V$ of minimal models CFTs. [3] In this case we have complete knowledge of the conformal data set $\{\Delta_i, C_{ij}^\phi\}$, where $C_{ij}^\phi$ are the Operator Product Expansion (OPE) coefficients.

The Hamiltonian Truncation approach consists in diagonalizing the finite dimensional matrix

$$H_{ij} \equiv \delta_{ij} \, \Delta_i / R + V_{ij} \quad, \qquad \Delta_i \leqslant \Delta_T \tag{3}$$

where we have defined $V_{ij} \equiv \langle O_i | V | O_j \rangle$. In this approach there is a numerical error due to the finite truncation energy cutoff $E_T \equiv \Delta_T / R$. The expectation, based on perturbation theory, is that for strongly relevant perturbations the spectrum of the truncated Hamiltonian converges to the QFT spectrum with a power like rate $\Delta_T^{-a}$.

Increasing $\Delta_T$ is computationally costly and therefore it is worth finding strategies to improve convergence without necessarily increasing $\Delta_T$. One such strategy is to use an Effective Hamiltonian [3–13]. The intuitive idea is to add EFT operators to (3), which act on the truncated Hilbert space, and take into account the effect of *integrating out* the higher energy eigenstates with $\Delta_i > \Delta_T$. The Effective Hamiltonian is not unique and many convenient choices have appeared in the literature, like for instance

$$H_{\text{eff}} = H + \Delta H_2 + \Delta H_3 + \ldots \tag{4}$$

where $(\Delta H_2)_{ij} = \frac{1}{2} V_{ik} (\frac{1}{E_{ik}} + \frac{1}{E_{jk}}) V_{kj}$, $(\Delta H_3)_{ij} = \frac{1}{2} \frac{V_{ik} V_{kk'} V_{k'j}}{E_{ik} E_{ik'}} - \frac{1}{2} \frac{V_{il} V_{lk} V_{kj}}{E_{ki} E_{kl}} + (i \leftrightarrow j)$, we defined $E_{ij} = \Delta_i / R - \Delta_j / R$, and a sum over the high energy states $\Delta_k, \Delta_{k'} > \Delta_T$ and low energy states $\Delta_l \leqslant \Delta_T$ is implicit – see the discussion in [12].

---

[2]For simplicity, in Sec. 1 we consider the case where $H_{\text{CFT}}$ is deformed by single relevant operator. In section 6 we discuss an example with multiple deformations $H_{\text{CFT}} + \sum_i V_i$ – the generalization to multiple deformations is straightforward.

[3]In $d = 2$, the spectrum of $H_{\text{CFT}}$ is shifted by $c/(12R)$, where $c$ is the central charge.

It turns out that Effective Hamiltonians have yet another useful application, besides improving the convergence rate as a function of $\Delta_T$ of the low energy spectrum. This application is in the context of QFTs that require renormalization. Next we summarize the main idea introduced in Ref. [12].

## QFTs requiring renormalization

If the deformation operator $\phi_\Delta$ in (1) has dimension $\Delta \geqslant d/2$, then the $n$-point correlation functions $\langle E_i | \phi_\Delta(x_1) \cdots \phi_\Delta(x_n) | E_j \rangle$ are generically [4] UV divergent when two or more $\phi_\Delta$'s approach each other. The spectrum of the QFT Hamiltonian is given by integrals of such $n$-point functions (see Ref. [14] for a review) and therefore one needs to carry out a renormalization procedure in order to define the QFT spectrum. For instance, the second order correction to the ground state energy is $E_{\text{g.s.}}^{(2)} \sim \int d^d x |x|^{\Delta-d} \langle \phi_\Delta(x) \phi_\Delta(1) \rangle = \int d^d x |x|^{\Delta-d} |x-1|^{-2\Delta}$, which diverges due to the singularity at $x \to 1$ for $\Delta \geqslant d/2$.

In order to define a UV finite theory of HT we will follow three simple steps:

1) Proceed first with standard QFT renormalization theory: introduce a short distance regulator $\epsilon$ (or any other local regulator such as momentum cutoff, or Pauli-Villars) and a finite set of counterterms to remove UV divergences from integrated correlation functions of the deformation $\phi_\Delta$. Schematically, (1) is replaced by

$$H_{\text{CFT}} + V + V_{\text{c.t.}}(\epsilon) \tag{5}$$

where $V_{\text{c.t.}}(\epsilon)$ is a set of integrals of local operators with divergent coefficients as $\epsilon \to 0$ that are engineered in order to define local and Lorentz invariant QFT correlation functions. The theory defined by (5) has an implicit dependence on $\epsilon$ (besides its explicit dependence through $V_{\text{c.t.}}(\epsilon)$) that arises through the use of the short distance regulator, that renders infinite sums over states in the QFT finite. However, the matrix elements of $H_{\text{CFT}}$ and $V$ between any specific pair of states are independent of $\epsilon$. The deformation $\phi_\Delta$ is a relevant operator ($\Delta < d$) and therefore only a finite number of counterterms need to be computed. Let us denote by $N$ the order of perturbation theory at which the counter-terms need to be computed to remove all UV divergences of integrated correlation functions of $\phi_\Delta$.

2) Next truncate the theory (5) to a low energy subsector spanned by the states $\{|E_i\rangle\}$ with energies $\Delta_i \leqslant \Delta_T$. [5] In principle one could use this truncated finite-dimensional matrix for Hamiltonian Truncation calculations. This approach would be difficult to implement because **(a)** it would require computing the matrix $(V_{\text{c.t.}}(\epsilon))_{ij}$ for a non-zero value of the local regulator $\epsilon$, and **(b)** one would need to perform two numerical extrapolations of the spectrum, extrapolating to $\Delta_T \to \infty$ first and then to $\epsilon \to 0$. To avoid this double extrapolation procedure, we compute the Effective Hamiltonian up to a perturbative order $N^* \geqslant N$. Schematically, we are led to the following Effective Hamiltonian

$$\delta_{ij} \Delta_i + V_{ij} + V_{\text{c.t.}}(\epsilon)_{ij} + H_{\text{eff},2}(\epsilon)_{ij} + \cdots + H_{\text{eff},N^*}(\epsilon)_{ij} \quad , \qquad \Delta_i \leqslant \Delta_T \tag{6}$$

All in all, *step 2 instructs*: truncate the Hamiltonian (5) and compute the Effective Hamiltonian $H_{\text{eff}}$ to the proper order.

---

[4] Barring the possibility of cancellations due to symmetries or selection rules.

[5] The cutoff $\Delta_T$ is a UV regulator. However it is inconvenient to use it to compute directly the counterterms because it does not preserve Lorentz invariance and breaks locality. See Ref. [14] for a detailed analysis of the breakdown of locality when this regulator is used.

3) Next, we observe that the matrix elements of (6) are finite in the limit $\epsilon \to 0$. In other words, the operator

$$K_{\text{eff}} \equiv \lim_{\epsilon \to 0} \left[ V_{\text{c.t.}}(\epsilon) + H_{\text{eff},2}(\epsilon) + \cdots + H_{\text{eff},N^*}(\epsilon) \right] \qquad (7)$$

is finite. We will therefore take this limit and use the Hamiltonian

$$H_{ij} \equiv \delta_{ij} + \Delta_i + V_{ij} + (K_{\text{eff}})_{ij} \quad , \qquad \Delta_i \leqslant \Delta_T \qquad (8)$$

The Hamiltonian (8) is the one that we advocate using for RG-flows that require renormalizing coupling constants. In order to get a finite spectrum it is enough to compute $H_{\text{eff}}$ to order $N^* = N$. Nevertheless it can be advantageous to compute higher orders of $H_{\text{eff}}$, i.e. $N^* > N$ to improve the rate of convergence of results with $\Delta_T$.

A couple of comments are in order.

In step 3) we are commuting two limits, by taking $\epsilon \to 0$ first, and then $\Delta_T \to \infty$, instead of performing $\Delta_T \to \infty$ first and then $\epsilon \to 0$. It is unclear to us why this would necessarily be a problem. Below we will numerically study a number of different RG flows using this methodology and we will find perfect agreement with the analytic expectations. The examples we consider here employ equal-time quantization, however this effective Hamiltonian approach can be applied if lightcone quantization [19] were used instead.

In (8) the matrix $(K_{\text{eff}})_{ij}$ effectively provides the counterterms needed if we had proceeded by regularizing UV divergences with the non-local cutoff $\Delta_T$. What we have achieved with our construction is a systematic way of computing such non-local counterterms by a procedure that makes apparent the scheme that is being used (which gets defined in step 1).

## 3   Selected models for study

The rest of this paper contains the main new results. We will present a number of Hamiltonian Truncation calculations of increasing difficulty, to demonstrate how to apply our procedure summarized in Sec. 2.

The first example we will study is the flow defined as the 2d critical Ising model (a.k.a. the $M(3,4)$ minimal model) deformed by the $\epsilon$ operator. This RG flow has a logarithmic divergence in its cosmological constant, and we use this simple setting to present a proof of concept of our method. We demonstrate the convergence of the spectrum after renormalization, and use $K_{\text{eff}}$ to accurately compute the Zamolodchikov $c$-function across a wide range of couplings. This flow has been recently studied in [20].

Next, we move on to a more challenging RG flow, from the tricritical Ising deformed by $\epsilon'$ to the critical Ising model ($M(4,5) + \phi_{1,3} \to M(3,4)$ in the minimal models' nomenclature). This RG flow has been previously studied through the TBA [21] as well as the TCSA with [3] and without [22] renormalization corrections. We will apply our systematic treatment of renormalization to this RG flow and we will demonstrate that we are able to get precise agreement for the expected Ising spectrum and central charge.

In both the Ising $+ \epsilon$ and Tricritical Ising $+ \epsilon'$, the only UV divergence is in the cosmological constant. Hamiltonian Truncation 'counterterms' provided by $K_{\text{eff}}$ are state dependent (non-local), and thus these computations provide a non-trivial test of our procedure. Moving forward we will present the study of an RG flow that requires coupling constant renormalization.

Table 1: A summary of the three strongly coupled RG flows we study in this work, where here 'ops.' refers to relevant operators. The dimension of the perturbing operator is indicated with a superscript. The counterterms generated by the $K_{\text{eff}}$ are indicated in schematic form. We have omitted its non-local structure and only highlighted local part that dominates for large $\Delta_T$.

| Flow | Bare Hamiltonian | Counterterms |
|---|---|---|
| 1. Ising + $\mathbb{Z}_2$-even ops. | $H^{\text{CFT}}_{\text{Ising}} + m\epsilon^{(1)}$ | $m^2 \log \Delta_T \mathbb{1}$ |
| 2. Tricritical to Critical Ising | $H^{\text{CFT}}_{\text{Tric. Ising}} + g\epsilon'^{(6/5)}$ | $g^2 \Delta_T^{2/5} \mathbb{1}$ |
| 3. $M(3,7)$ + $\mathbb{Z}_2$-even ops. | $H^{\text{CFT}}_{M(3,7)} + ig_3 \phi_{1,3}^{(-2/7)} + g_5 \phi_{1,5}^{(8/7)}$ | $g_5^2 \Delta_T^{2/7} \mathbb{1} + g_5^2 \Delta_T^{4/7} \phi_{1,3}$ |

The third QFT we will study is the $M(3,7)$ minimal model deformed by two of its relevant operators $\phi_{1,3}$ and $\phi_{1,5}$. This QFT is one of the simplest RG flows requiring non-trivial UV renormalization of operators. The Hilbert space we work with is spanned by only the three primary operators $\mathbb{1}$, $\phi_{1,3}$ and $\phi_{1,5}$. The only UV divergences appear at second order in the $\phi_{1,5}$ coupling. These divergences are taken care of by introducing counterterms proportional to $\mathbb{1}$ and $\phi_{1,3}$. This model involves only one extra operator to be renormalized w.r.t. the previous examples.

In Tab. 1 we show a summary of the three theories we study, the UV divergences present in each RG flow and their dependence as a function of the cutoff $\Delta_T$. The common thread in the three flows we study is that the UV divergences arise only at second order in the coupling. This is because in the three flows the scaling dimensions of the deforming operators satisfy $1 \leqslant \Delta < 4/3$.

## 3.1 Hamiltonian Truncation for Minimal Models

In this section we adapt the general discussion in Sec. 2 to the case of QFTs defined by deforming $d = 2$ minimal models on the cylinder $\mathbb{R} \times S_R^1$. This section is mostly review and will serve to define our notation.

We use cylinder coordinates that are defined as $(\tau, x) = R(\log r, \theta)$, where $(r, \theta)$ are the $\mathbb{R}^2$ polar coordinates. Here $\tau \in \mathbb{R}$ is the time along the un-compact cylinder direction, while $x \in [0, 2\pi R]$ is the compactified spatial coordinate. On this geometry, the CFT Hamiltonian $H_{\text{CFT}}$ can be written in terms of the Virasoro generators of the CFT on the plane:

$$H_{\text{CFT}} = \frac{1}{R}\left(L_0 + \bar{L}_0 - \frac{c}{12}\right), \tag{9}$$

where $c$ is the central charge of the CFT. The full QFT Hamiltonian instead reads:

$$H = H_{\text{CFT}} + \sum_i \frac{g_i}{2\pi} \int_0^{2\pi R} dx\, \phi_{\Delta_i}(0, x), \tag{10}$$

where the $g_i$'s are dimensionful coupling constants of dimension $2-\Delta_i$ and the $\phi_{\Delta_i}$ are relevant primary scalar fields of dimension $\Delta_i$, evaluated at zero cylinder time.

The basis of states that we use to compute the matrix elements of the Hamiltonian are made up of a string of Virasoro modes acting on primary states:

$$|\psi\rangle = L_{-n_1}\dots L_{-n_k}\bar{L}_{-m_1}\dots\bar{L}_{-m_l}|h\rangle, \quad n_1 \geqslant \dots \geqslant n_k > 0, m_1 \geqslant \dots \geqslant m_l > 0, \tag{11}$$

with $|h\rangle = \phi_{r,s}(0,0)|0\rangle$, where $(r,s)$ indices refer to the Kac table position. [6] The Hamiltonian in (10) conserves spatial momentum, and we choose to work in the zero momentum sector, where $\sum_{i=1}^{k} n_i = \sum_{j=1}^{l} m_j$. These states are eigenstates of the CFT Hamiltonian (9), with eigenvalue given by $\frac{1}{R}(2h + 2\sum_i n_i - c/12) = \frac{1}{R}(\Delta_\psi - c/12)$. Then, the matrix elements of a deformation term $V_j \equiv \frac{g_j}{2\pi} \int_0^{2\pi R} dx\, \phi_{\Delta_j}(0,x)$ between two states of the form (11) are given by

$$\langle \psi_f | V_j | \psi_i \rangle = g_j R \langle \psi_f | \phi_{\Delta_j}(0,0) | \psi_i \rangle = g_j R^{1-\Delta_j} \tilde{C}_{fi}^{j}. \tag{12}$$

where $\tilde{C}_{fi}^{j}$ refers to the dimensionless three point function $\lim_{|x_f| \to \infty} |x_f|^{2\Delta_f} \langle \mathcal{O}_f(x_f) \phi_{\Delta_j}(1) \mathcal{O}_i(0) \rangle$, where $\phi_{\Delta_j}$ is the plane primary field evaluated at an arbitrary point on the unit circle of the plane. The coefficients $\tilde{C}_{fi}^{j}$ are completely fixed by conformal symmetry in terms of the CFT data $\{\Delta_i, C_{jk}^{i}\}$ and can be computed numerically. When $\mathcal{O}_f, \mathcal{O}_i$ are primary operators, $\tilde{C}_{fi}^{j}$ is of course simply the OPE coefficient $C_{fi}^{j}$. The three-point functions can be factorised into holomorphic and anti-holomorphic parts. We calculate the two parts recursively before combining them - a similar method is described in Ref. [23]. We developed Mathematica and Python codes to compute and crosscheck our numerical results, which are available upon request.

Summarizing, the matrix elements of the Hamiltonian (10) between our basis states are given by:

$$\langle \psi_f | H | \psi_i \rangle = \delta_{\Delta_f, \Delta_i} \langle \psi_f | \psi_i \rangle \frac{1}{R}\left(\Delta_i - \frac{c}{12}\right) + \sum_j g_j R^{1-\Delta_j} \tilde{C}_{fi}^{j}. \tag{13}$$

The basis $\{|\psi_i\rangle\}$ we'll be working with is not orthonormal, and thus we deal with the generalized eigenvalue problem $H\vec{v} = EG\vec{v}$, where $G_{ij} = \langle \psi_i | \psi_j \rangle$ is the non-singular [7] Gram matrix.

We will now apply the procedure reviewed in Sec. 2 on several examples to compute the counter-terms for Hamiltonian Truncation. We start by applying our formalism to two RG-flows that have been well studied in the literature. The following two sections will serve as a crosscheck and to gear up for more involved studies.

# 4 Example I: Ising + $\epsilon$

In this section we consider the 2d Ising CFT deformed by its least relevant primary operator, $\epsilon(x)$. This QFT is closely related to the free massive fermion, and hence it is solvable. In Subsec. 4.1 we describe the Ising CFT, the UV divergences, and give the bare Hamiltonian as well as the counterterm computed from the Effective Hamiltonian procedure. In Subsec. 4.2 we demonstrate the convergence of the spectrum, and we accurately reproduce the Zamolodchikov $c$-function for a range of values of the coupling.

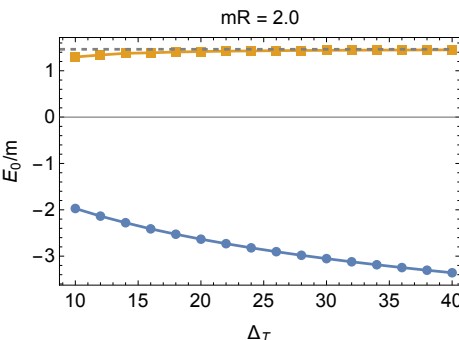
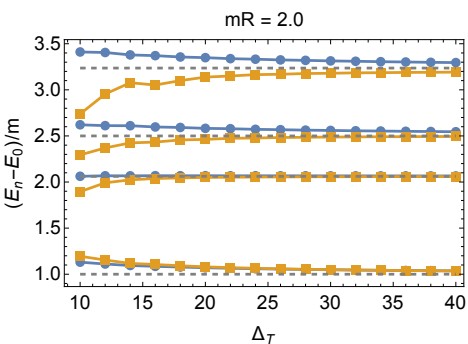

Figure 1: Plots indicating the variation of Hamiltonian Truncation estimates for the Ising + $\epsilon$ spectrum with the truncation parameter $\Delta_T$. Results obtained using the bare Hamiltonian (14) are shown in blue and estimates using (17) are shown in orange. The exact results for the spectrum of this solvable QFT are plotted using dashed gray lines. Further details are provided in the text.

## 4.1 The Hamiltonian Truncation matrix

The Hamiltonian of the Ising CFT deformed by $\epsilon$ is given by:

$$H_{\text{Ising}} + V = \frac{1}{R}\left(L_0 + \bar{L}_0 - \frac{c_{\text{Ising}}}{12}\right) + \frac{m}{2\pi}\int_0^{2\pi R} dx\, \epsilon(0, x)\,, \tag{14}$$

where without lost of generality the field $\epsilon(0, x)$ is evaluated at zero cylinder time.

The central charge is $c_{\text{Ising}} = \frac{1}{2}$. The local operators of the CFT are given by three primary states $\mathbb{1}, \epsilon, \sigma$ (of scaling dimensions respectively 0, 1 and 1/8) and their Virasoro descendants. The fusion rules are given by

$$\epsilon \times \epsilon = \mathbb{1}\,, \quad \sigma \times \epsilon = \sigma\,, \quad \sigma \times \sigma = 1 + \epsilon\,. \tag{15}$$

The CFT has a global $\mathbb{Z}_2$ symmetry under which $\mathbb{1}$ and $\epsilon$ are even and $\sigma$ is odd; as well as Krammers-Wannier self-duality which maps $\epsilon \to -\epsilon$. The only non-trivial OPE coefficient is $C_{\sigma\sigma\epsilon} = 1/2$.

The second order correction to the CFT spectrum is given by the integral over the cylinder of the two-point function $\langle i|\epsilon(x)\epsilon(0)|i\rangle$. Because the operator $\epsilon(x)$ has a scaling dimension one, this correlation function has a singularity of $\sim 1/|x|^2$ as $x$ approaches zero, giving the integral a logarithmic UV divergence.

In order to deal with this divergence in Hamiltonian Truncation, we next perform the steps described in Sec. 2: we first add a local counterterm (proportional to the identity) to the $\epsilon$-regulated theory, then compute the Effective Hamiltonian up to second order in the coupling $m$ and finally remove the local regulator by taking the $\epsilon \to 0$ limit. We are led to the following $K_{\text{eff},2}$ operator:

$$\left(K_{\text{eff, Ising}}\right)_{fi} = \langle f|i\rangle\, \frac{m^2 R}{2}\left[\sum_{n=0}^{2n\leqslant\Delta'_{T,i}} + \sum_{n=0}^{2n\leqslant\Delta'_{T,f}}\right]\frac{1}{2n+1}\,, \tag{16}$$

---

[6] Because we are considering diagonal modular invariant partition functions there are no non-scalar primaries. Thus the $\phi_{r,s}$ has the same holomorphic $h$ and anti-holomorphic $\bar{h} = h$ dimension, and total scaling dimension $2h$.

[7] We construct our basis iteratively, adding Virasoro descendants with the structure indicated in (11) one by one and accepting the new state only if the Gram matrix has no zero eigenvalues. This ensures our truncated basis is free of Virasoro null states with zero norm.

where for this flow, we define $\Delta'_{T,j} \equiv \Delta_T - 1 - \Delta_j$. The operator (16) needs to be added to (14) in order to get a finite spectrum, determined by our choice of local regulation scheme. We remark that the non-locality of $K_{\text{eff, Ising}}$ is engineered such that, together with the non-local Hamiltonian Truncation cutoff, it leads to a spectrum that is compatible with a local QFT. We provide the detailed calculation of (16) in appendix A.

All in all, the truncated Hamiltonian matrix of the Ising+$\epsilon$ QFT is given by

$$H_{ij} = (H_{\text{Ising}})_{ij} + V_{ij} + \left(K_{\text{eff, Ising}}\right)_{ij} \quad \text{with} \quad \{\Delta_i, \Delta_j\} \leqslant \Delta_T . \tag{17}$$

## 4.2 Numerical results

In the left panel of Fig. 1, we plot the ground state energy at moderate coupling $mR = 2.0$, estimated using HT with different values of $\Delta_T$. In blue, we show the bare HT result - taken without using $K_{\text{eff, Ising}}$. As expected, this estimate diverges, growing logarithmically with $\Delta_T$. In orange, we plot HT results using the full second-order Effective Hamiltonian shown in (17), which indicate convergence to a finite value as $\Delta_T$ is increased. For $\Delta_T = 40$, the truncated Hilbert space has dimension 22,127.

Using the massive fermion description, we can compute the ground state energy analytically for our choice of renormalization scheme. The analytic result $E_0/m = 1.464\ldots$ is plotted on the figure as the dashed gray line. The derivation is presented in appendix B. We see that the HT estimate in orange is converging towards this value.

In the right panel of Fig. 1, we plot energy differences between the ground state and first four excited states for different values of the HT truncation parameter. In this case, both the bare HT and Effective Hamiltonian computations converge to the exact analytic results (the gray dashed lines) as $\Delta_T$ is increased. For the two highest excited states, the Effective Hamiltonian computation provides a more accurate estimate at finite $\Delta_T$. The first four excited states correspond to a state with a single massive fermion, two states with a pair of fermions with equal and opposite momenta and one three fermion state. The four energy differences are given by $\Delta E/m = 1, \sqrt{17}/2, 5/2$ and $1 + \sqrt{5}$ respectively. See appendix B for further explanation.

In (17), we only evaluated the Effective Hamiltonian up to second order, and neglected contributions of order $O(m^3)$. The largest missing terms come from the $\epsilon$ contribution to $H_{\text{eff 3}}$ and the identity contribution to $H_{\text{eff 4}}$. For large $\Delta_T$, they scale as $\sim \Delta_T{}^p$ where $p = 3\Delta - 2d - \Delta = -2$ in the $H_{\text{eff 3}}$ case, and $p = 4\Delta - 3d = -2$ also in the $H_{\text{eff 4}}$ case.

In Fig. 2, we show the variation of the ground state energy with the coupling parameter $mR$. We compare HT calculations obtained using (17) with the exact result derived in appendix B. The ground state energy eigenvalue has a logarithmic dependence on the radius, not present in the matrix elements, that emerges upon diagonalisation. For small values of $mR$, there is precise agreement between the HT and exact results. However for larger $mR$, HT estimates (at finite $\Delta_T$) suffer from larger systematic errors due to the omission of higher-order corrections to the Effective Hamiltonian. As a result, a gap emerges between the exact and HT results at the top of the $mR$ range.

For unitary 1+1 dimensional QFTs, the Zamolodchikov $c$-function [24] is a monotonically increasing function of energy scale, which interpolates between the central charges of two CFTs, which describe the QFT in its high and low energy limits. By calculating this function $c(\mu)$, and checking that it interpolates between the central charges of the empty and Ising CFTs, we get another consistency check of our HT method.

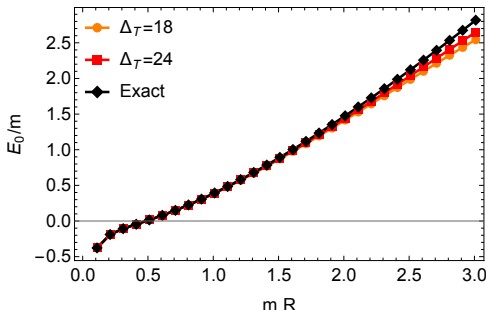

Figure 2: The variation of the ground state energy with coupling $mR$ in the Ising+$\epsilon$ QFT. In black is the exact analytic result given by (B.6). The HT estimates, taken using the second-order Effective Hamiltonian shown in (17) are plotted in orange and red, for truncation levels $\Delta_T = 18, 24$ respectively.

The Zamolodchikov-$c$ function has the spectral representation

$$\Delta c(\mu) \equiv c(\mu) - c_{\text{IR}} = 12\pi \lim_{s_{min} \to 0} \int_{s_{min}}^{\mu^2} \frac{ds}{s^2} \rho_\Theta(s), \tag{18}$$

see for example [25]. In HT, the spectral density of the stress tensor trace $\rho_\Theta$ is given by the sum over delta functions below, since the spectrum is discrete

$$\rho_\Theta(\mu^2) = 4\pi R \sum_{i \geqslant 1} \mu_i \delta(\mu^2 - \mu_i^2) |\langle \Omega | \Theta(0) | E_i \rangle|^2 . \tag{19}$$

Here $\mu_i \equiv E_i - E_0$, $\langle E_i | E_j \rangle = \delta_{ij}$ and $E_0$ is the energy of the interacting theory vacuum state $|\Omega\rangle$. Plugging (19) into (18) yields [20]

$$\Delta c(\mu) = 48\pi^2 R \sum_{i=1}^{\mu_i \leqslant \mu} \frac{|\langle \Omega | \Theta(0) | E_i \rangle|^2}{\mu_i^3} . \tag{20}$$

We compute the sum above using HT. If the only UV divergences are in the cosmological constant, like in the flow under consideration, all the terms of (20) are finite.

The matrix elements of the stress tensor trace between zero momentum eigenstates used in (20) are independent of the spatial coordinate, and may be calculated using

$$\Theta = \frac{1}{R} \frac{\partial}{\partial R} \left[ R H_{\text{eff}}(R) \right] , \tag{21}$$

where the partial derivative is taken with the dimensionful coupling $m$ fixed in (14) and (16). The formula above captures not only the contribution from the relevant deformation $V$, but also corrections to the operator arising from modes above the HT cutoff scale $\Delta_T$, which are encoded in the higher order terms of the Effective Hamiltonian. Corrections to operators that come when an Effective Hamiltonian is used are discussed in [12].

In Fig. 3, we evaluate the sum in (20) for different values of the parameter $mR$. In the infinite volume limit $mR \to \infty$, we expect the $c$-function to interpolate between 0 and 1/2 (the trivial and Ising CFT central charges) following the curve $c(\mu) = 1/2(1 - 4m^2/\mu^2)^{3/2}$, which is plotted in black. We normalize the horizontal axis using $m_{\text{gap}} = E_1 - E_0$.

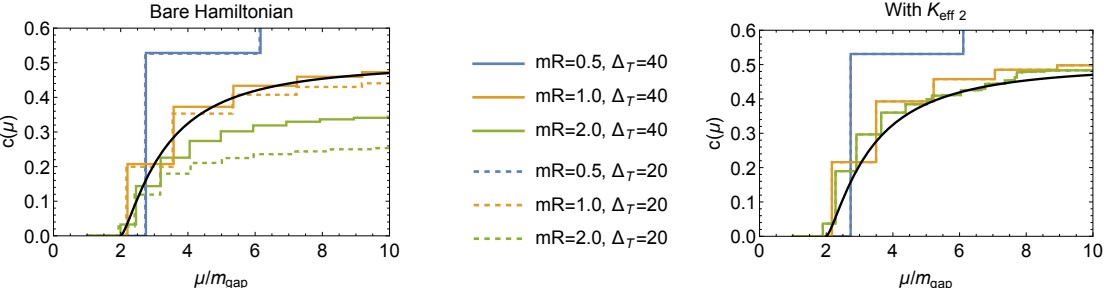

Figure 3: In the left panel we show plots of the Zamolodchikov c-function, produced using the bare Hamiltonian (14). Different colors refer to the different values of $mR = 0.5$, 1 and 2. Dashed lines refer to lower truncation level. The analytic result for $c(\mu)$ along this flow is plotted as the solid black line. In the right panel, we plot the same functions using (17) and (21).

In the left panel, we plot results obtained using the bare Hamiltonian in Eq. (14). Our result for this quantity agrees with the earlier results of Ref. [20]: although the HT result for $mR = 1.0$ approximates the infinite volume expectation well, the result at $mR = 0.5$ shows a large finite volume correction, and the curve at $mR = 2.0$ does not match the expected result either. In the latter case, there is a large visible difference between the HT data taken using $\Delta_T = 20$ and $\Delta_T = 40$, suggesting that systematic errors due to the finite truncation level are large.

In the right panel, we plot results obtained using the second-order Effective Hamiltonian from (17). The data at $mR = 2.0$ is much better converged in $\Delta_T$, and now lines up more closely with the analytic infinite volume result.

In Ref. [20], it was observed that the computation of the Zamolodchikov $c$-function in a boosted frame led to better converged results. This observation provided the basis for an intuitive comparison with the conformal light-cone computations. Fig. 3 reveals that a similar improvement on the calculation of the Zamolodchikov $c$-function is possible thanks to the Effective Hamiltonian corrections.

## 5 Example II: Tricritial Ising + $\epsilon'$

In this section we study the flow generated by deforming the Tricritical Ising CFT with the $\phi_{1,3}$ operator. In Subsec. 5.1, we describe the Tricritical Ising CFT and give the form of the bare Hamiltonian as well as the Effective Hamiltonian corrections which renormalize the truncated theory. In Subsec. 5.2 we demonstrate the convergence of the spectrum and analyze the various improvement terms added to the bare Hamiltonian. Finally, in Subsec. 5.3 we interpret the HT data in the IR with the Ising EFT. We conclude by computing the Zamolodchikov $c$-function for a range of couplings using the Effective Hamiltonian.

### 5.1 The Hamiltonian Truncation matrix

The primary operators of the Tricritical Ising CFT, as well as their scaling dimensions and charges under a global $\mathbb{Z}_2$ and Kramers-Wannier self-duality symmetry are summarized in Table 2.

Table 2: The 6 primaries of the Tricritical Ising model (with the conventional names given to the operators indicated in parentheses), along with their scaling dimensions and transformation properties under a global $\mathbb{Z}_2$ and the Kramers-Wannier duality.

| Tricritical Ising primaries | $\phi_{1,1}(\mathbb{1})$ | $\phi_{1,3}(\epsilon')$ | $\phi_{1,2}(\epsilon)$ | $\phi_{1,4}(\epsilon'')$ | $\phi_{2,2}(\sigma)$ | $\phi_{2,1}(\sigma')$ |
|---|---|---|---|---|---|---|
| $\Delta$ | 0 | 6/5 | 1/5 | 3 | 3/40 | 7/8 |
| $\mathbb{Z}_2$ | Even | Even | Even | Even | Odd | Odd |
| KW | Even | Even | Odd | Odd | $\mu$ | $\mu'$ |

The Hamiltonian reads:

$$H_0 + V = \frac{1}{R}\left(L_0 + \bar{L}_0 - \frac{c_{4,5}}{12}\right) + \frac{g}{2\pi}\int_0^{2\pi R} dx\, \epsilon'(0,x), \tag{22}$$

where the central charge is $c_{4,5} = 7/10$. The fusion rules for the primary operators listed in Table. 2, while the OPE coefficients are listed in appendix C.

Similarly to the Ising+$\epsilon$ theory studied in the previous section, there is one UV divergence in this QFT, appearing at second order in perturbation theory. This perturbative correction to the energy of state $|i\rangle$ is given by the integrated two point function $\int d^2x\, \langle i|\epsilon'(x)\epsilon'(0)|i\rangle$. It is a power-like UV divergence because $2\Delta_{\epsilon'} - 2 > 0$ (see e.g. [14] for a review of conformal perturbation theory). Again, only the identity operator requires renormalization in this QFT.

We go through the procedure of Hamiltonian Truncation renormalization described in Sec. 2 and we are led to

$$\left(K_{\text{eff},2}\right)_{fi} = \langle f|i\rangle \frac{g^2 R^{3/5}}{2}\left[\sum_{n=0}^{2n\leqslant\Delta'_{T,i}} + \sum_{n=0}^{2n\leqslant\Delta'_{T,f}}\right]\frac{1}{2n+6/5}\left(\frac{\Gamma(n+6/5)}{\Gamma(n+1)\Gamma(6/5)}\right)^2 \tag{23}$$
$$+ \langle f|\epsilon'(1)|i\rangle\, C^{\epsilon'}_{\epsilon'\epsilon'}\frac{g^2 R^{3/5}}{2}\left[\sum_{2n>\Delta'_{T,i}} + \sum_{2n>\Delta'_{T,f}}\right]\frac{1}{2n+6/5}\left(\frac{\Gamma(n+3/5)}{\Gamma(n+1)\Gamma(3/5)}\right)^2 .$$

For this model, we define $\Delta'_{T,j} = \Delta_T - 6/5 - \Delta_j$. We then obtain a truncated Hamiltonian for our QFT

$$H_{ij} = (H_0)_{ij} + V_{ij} + \left(K_{\text{eff},2}\right)_{ij} \quad \text{with} \quad \{\Delta_i, \Delta_j\} \leqslant \Delta_T . \tag{24}$$

Besides including the needed counterterms at second order, we also include UV finite terms to improve the rate of convergence of the spectrum with increasing cutoff $\Delta_T$ – they play the same role as the improvement terms used in [3, 4].

To do this, we include from $H_{\text{eff},2}$ the contribution from the $\epsilon'$ primary appearing in the $\epsilon' \times \epsilon'$ OPE (shown in the second line of (23)), while from $H_{\text{eff},3}$ we include the leading contribution from the identity operator. We obtain the following expression for the $K_{\text{eff}}$ operator at third order in $g$:

$$\left(K_{\text{eff},3}\right)_{fi} = \langle f|i\rangle\, C^{\epsilon'}_{\epsilon'\epsilon'}\frac{g^3 R^{7/5}}{2}\left[\left(\sum_{x,y>\Delta'_{T,i}/2} + \sum_{x,y>\Delta'_{T,f}/2}\right)\Xi_{\text{c}}(y,x) + \left(\sum_{\substack{x=0\\2y>\Delta'_{T,i}}}^{2x\leqslant\Delta'_{T,i}} + \sum_{\substack{x=0\\2y>\Delta'_{T,f}}}^{2x\leqslant\Delta'_{T,f}}\right)\Xi_{\text{d}}(y,x)\right]. \tag{25}$$

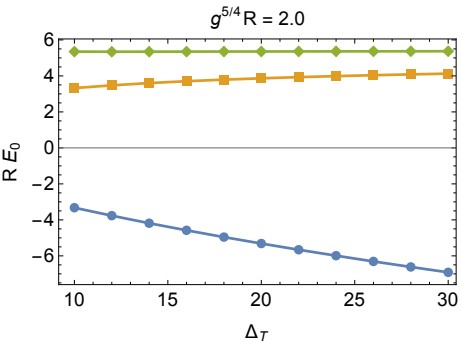 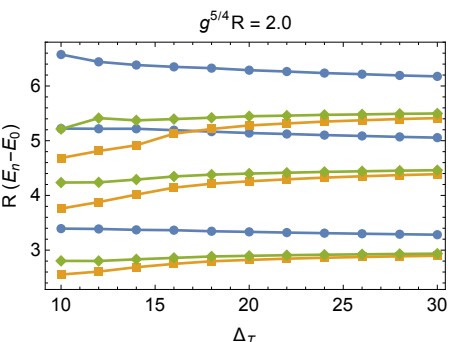

Figure 4: Plots indicating the variation of the Tricritical Ising $+\epsilon'$ ground state energy (left panel) and three energy gaps (right panel) with the truncation parameter $\Delta_T$. The gaps plotted are the differences between the three smallest excited state energies, and the ground state energy in the subsector of the QFT that is even under both $\mathbb{Z}_2$ and KW discrete symmetries. For $\Delta_T = 30$, this subsector contains a total of 12397 independent states. The bare Hamiltonian estimates are shown in blue, the estimates using $K_{\text{eff},2}$ are shown in orange and the estimates using $K_{\text{eff},2} + K_{\text{eff},3}$ are shown in green.

The connected and disconnected summands are given by

$$\Xi_c(y,x) \equiv \frac{\Gamma^2\left(\frac{1}{5}(3+5x)\right)\Gamma^2\left(\frac{1}{5}(3+5y)\right){}_3F_2(3/5,-y,-x;2/5-y,2/5-x;1)}{(2x+6/5)(2y+6/5)\Gamma^4(3/5)\Gamma^2(1+x)\Gamma^2(1+y)} , \tag{26}$$

$$\Xi_d(y,x) \equiv \frac{\Gamma^2\left(\frac{1}{5}(3+5x)\right)\Gamma^2\left(\frac{1}{5}(3+5y)\right){}_3F_2(3/5,-y,-x;2/5-y,2/5-x;1)}{(2x+6/5)(2x-2y)\Gamma^4(3/5)\Gamma^2(1+x)\Gamma^2(1+y)} . \tag{27}$$

Details on the calculation of these expressions are provided in appendix A. Once the improvement terms up to third order in the coupling $g$ are included, the truncated Hamiltonian is given by

$$H_{ij} + \left(K_{\text{eff},3}\right)_{ij}, \tag{28}$$

with $H$ given in (24).

## 5.2 Cutoff dependence

In Fig. 4 we show eigenvalues of the bare Hamiltonian $(H_0)_{ij} + V_{ij}$, the renormalized Hamiltonian (24), and the improved Hamiltonian (28), as a function of the truncation cutoff. In order to interpret the results we note that at large values of the truncation cutoff, the most significant terms in the Effective Hamiltonian scale as

$$K_{\text{eff},2} + K_{\text{eff},3} \sim g^2 \Delta_T^{2/5}\mathbb{1} + C_{\epsilon'\epsilon'}^{\epsilon'}g^2\Delta_T^{-4/5}\epsilon' + C_{\epsilon'\epsilon'}^{\epsilon'}g^3\Delta_T^{-2/5}\mathbb{1} + \mathcal{O}(\Delta_T^{-6/5}). \tag{29}$$

From Eq. (29), we can see that the improvement term at third order $H_{\text{eff},3}$ decays slower than the decaying term at second order, in the $\Delta_T \to \infty$ limit. This suggests that the Hamiltonian Truncation data computed with (28) should show an improved convergence rate with respect to the spectrum computed with (24). The largest subleading piece that we omit in (29) comes from the identity operator part of $H_{\text{eff},4}$, with a scaling of $\Delta_T$ to the power $4\Delta_{\epsilon'} - 3d = -6/5$.

The above expectations are indeed realized in Fig. 4. On the left plot we see that the vacuum energy diverges (blue) unless counterterms are added (green and orange). We also

see an improvement on convergence of the green curves with respect to the orange curves, thanks to the addition of the third order improvement terms. Although in Fig. 4 we only show the convergence of the even-even-sector eigenvalues, we find similarly well converged results in the other symmetry sectors.

Due to the small variation with the cutoff of the first few state energies, we conclude that systematic errors from using finite $\Delta_T$ are small for the numerical data taken using (28) for $\Delta_T = 30$, and we proceed with the physics. We next analyze and interpret our data using an Effective Field Theory (EFT). In order to do so we use the data obtained with the largest available $\Delta_T$ (i.e. no extrapolations to the infinite cutoff limit), and neglect any remaining truncation errors.

## 5.3 EFT interpretation

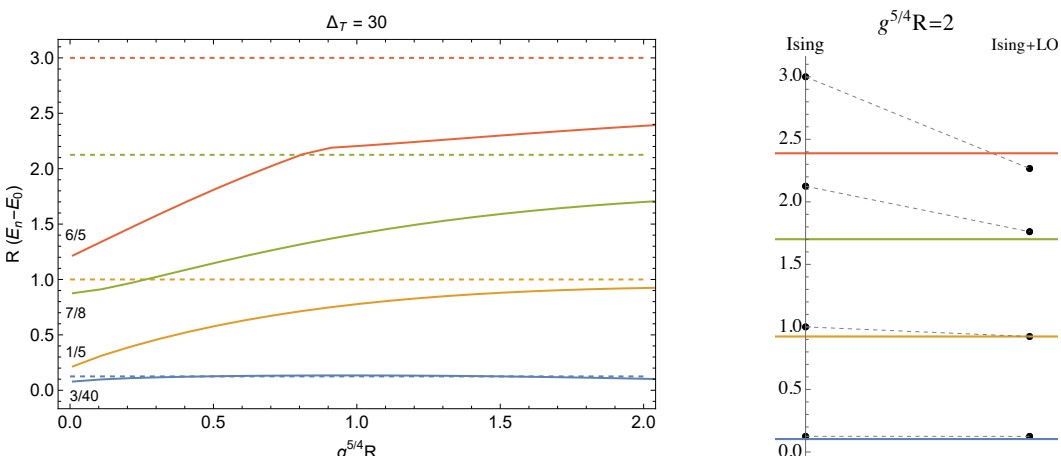

Figure 5: Left: the four smallest energy gaps along the Tricritical Ising flow. Results are obtained using the third order Effective Hamiltonian (28). The truncation parameter is $\Delta_T = 30$, corresponding to a truncated Hilbert space containing 40818 states in total (once all symmetry subsectors are included). The horizontal dashed lines correspond to the scaling dimensions of the $\sigma$, $\epsilon$, $L_{-1}\bar{L}_{-1}\sigma$ and $L_{-1}\bar{L}_{-1}\epsilon$ operators in the Ising model. Right: EFT fit. Horizontal lines are the numerical data from the left plot at fixed radius, following the same color code. The first column of four black dots are the lowest scaling dimensions of the Ising CFT and the right column of black dots are the result of the leading order fit described in the text. The dashed gray line is there to guide the eye.

In the left panel of Fig. 5, we show the spectrum of (28) as a function of the coupling of the $\epsilon'$ perturbation for $\Delta_T = 30$. As argued earlier, for this truncation value the low lying spectrum is well converged. At zero value of the coupling, the spectrum dimensions exactly match the scaling dimensions of the Tricritical Ising (shown in black next to each line). At large values of the coupling, in the IR, the spectrum is supposed to flow to that of the Ising CFT [21]. Indeed we see that the lowest two values show good agreement with the $\Delta_\sigma = 1/8$ and $\Delta_\epsilon = 1$ scaling dimensions of the Ising CFT. The highest energy levels are somewhat farther away from the assumed IR CFT values $\Delta_\sigma + 2$ and $\Delta_\epsilon + 2$. [8] We can interpret these deviations using an EFT

---

[8]We note that at $g^{5/4}R \approx 0.8-0.9$ the highest energy level being plotted shows a kink; this is due to the energy of the $|\epsilon'\rangle$ UV state crossing the $|L_{-1}\bar{L}_{-1}\epsilon\rangle$ UV state.

analysis. [9] The EFT action in the IR is given by

$$S_{\text{Ising}} + \int d^2x \left[ \Lambda^2 + \frac{g_{T\bar{T}}}{2\pi M^2} T^{\mu\nu} T_{\mu\nu}(x) + O(M^{-6}) \right], \tag{30}$$

where $\Lambda^2$ is the cosmological constant. The leading scalar irrelevant operator is $T^{\mu\nu}T_{\mu\nu}(x) \sim (L_{-2}\bar{L}_{-2}\mathbb{1})(x)$ and has dimension 4. The next scalar operator that we can write in the action has dimension eight,[10] and is consequently much more suppressed at low energies. We therefore expect that a leading order fit to the spectrum using (30) should be a good approximation.

From (30) we compute the finite volume spectrum,

$$E_i^{\text{EFT}} = R \left[ \Lambda^2 + \frac{E_i}{R} + g_{T\bar{T}} \frac{(E_i/2)^2}{(MR)^2} + O(g_{T\bar{T}}^2/M^4) + O(M^{-6}) \right] \tag{31}$$

where $E_i = (\Delta_i - c_{\text{Ising}}/12)/R$ are the energy levels of the Ising on the cylinder. The last equation was obtained by applying tree-level perturbation theory on (30). This equation was used in Ref. [21] to obtain the corrections in the Ising EFT by comparing with the Thermodynamic Bethe Ansatz expanded for large radius. After fixing units of the EFT cutoff to $M^2 = 1$, the gaps at leading order in the $g_{T\bar{T}}$ coupling are simply given by

$$\delta E_i^{\text{EFT}} \equiv (E_i^{\text{EFT}} - E_0^{\text{EFT}})R = \Delta_i + g_{T\bar{T}} \left[ (\Delta_i/2 - c_{\text{Ising}}/24)^2 - (c_{\text{Ising}}/24)^2 \right]. \tag{32}$$

Next we minimize the function $\chi^2(g_{T\bar{T}}) = \sum_{i=1}^{2} [\delta E_i^{\text{HT}} - \delta E_i^{\text{EFT}}(g_{\bar{T}T})]^2$ to estimate the EFT coupling $g_{\bar{T}T}$. Here $\delta E_i^{\text{HT}}$ are the values of the four lowest energy gaps shown in the left plot of Fig. 5. We show the result of the fit in the right plot of Fig. 5, for a given value of $g^{5/4}R$. The horizontal lines of the right plot are the energy levels of the left plot for a fixed values of $g^{5/4}R = 2$. The left-most black dots are the Ising scaling dimensions. Finally the rightmost points are the result of the fit, that is the value given by (32) with $g_{T\bar{T}}^* = -0.335$. We see that a simple one-parameter fit of the first two gaps is enough to greatly reduce the discrepancy between the four lowest energy-levels of the HT data and the IR CFT scaling dimensions. In this work we will content ourselves with this leading order fit. Higher order EFT fits may enable more precise quantitative agreement with the HT data and reveal more information about the IR physics, such as the scaling of the EFT Wilson coefficients as a function of the radius $R$ [27].

In Fig. 6, we plot the variation in the Zamolodchikov $c$-function for the Tricritical Ising CFT deformed with its $\epsilon'$ operator, evaluated using (20). We expect this function to increase by an amount equal to the difference between the central charges of the Tricritical Ising and Ising CFTs $c_{4,5} - c_{\text{Ising}} = 0.7 - 0.5 = 0.2$ as the energy scale $\mu$ is increased.

In the left panel of Fig. 6, we show results obtained for $\Delta c(\mu)$ using the bare Hamiltonian in (22) for a range of values of the dimensionless coupling or volume parameter $g^{5/4}R$. The differences between the curves' endpoints cover a wide range of values below 0.2. However for the most strongly coupled green curves at $g^{5/4}R = 1.5$, there is a sizable variation between the results for $\Delta_T = 16$ and $\Delta_T = 30$, indicating that truncation error may be large for these curves.

---

[9]Recently Ref. [26] performed a similar fitting analysis of Hamiltonian Truncation data for the Ising to Lee-Yang RG flow.

[10]The Tricritical Ising operator $\epsilon'$ is even under a global $\mathbb{Z}_2$ and is also invariant under the Kramers-Wannier self-duality of the CFT. These two symmetries explain the absence of EFT scalar operators in the $\sigma$ and $\epsilon$ family of the Ising CFT in (30).

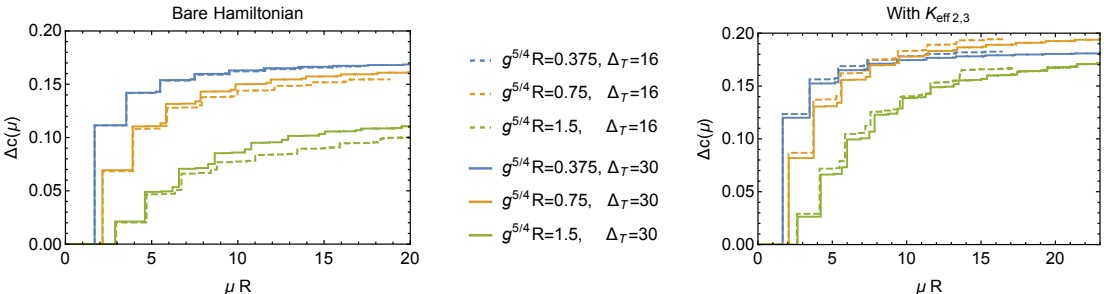

Figure 6: In the left panel we show plots of the Zamolodchikov c-function, produced using the bare Hamiltonian. Different colors refer to the different values of $g^{5/4}R = 0.375, 0.75, 1.5$. Dashed lines refer to lower truncation level. In the right panel, we plot the same functions using the Effective Hamiltonian in (28).

In the right panel, we plot the $c$-function evaluated using the Effective Hamiltonian from (28). In this case, there is much more consistency between the determinations of the endpoints made using different values of $g^{5/4}R$, and smaller differences between the $\Delta_T = 16$ and $\Delta_T = 30$ curves. The data shown is consistent with our expectation that $c$ should increase by 0.2 for large volume.

## 6 RG flows with coupling constant renormalization in HT

In this section, we will apply our Effective Hamiltonian formalism in the context of an RG-flow involving coupling constant renormalization. Our motivation stems from the fact that in higher dimensions $d > 2$ many RG flows of interest require renormalization of coupling constants. Therefore, besides their fascinating nature, non-perturbative $d = 2$ RG flows that involve UV divergences are of interest as a playground to understand conceptual problems in a simpler numerical set up than the higher dimensional case.

In order to carry this crosscheck of our formalism we will study the $M(3,7)$ 2d CFT, a non unitary minimal model, deformed by two of its relevant primary operators: $\phi_{1,3}$ and $\phi_{1,5}$ (the subscripts labeling the operators refer to their position in the Kac table). Out of the infinite set of minimal models (both unitary and non unitary), why did we choose the $M(3,7)$ as a case study?

Firstly, this model appears us relatively simple: due to symmetry selection rules, we can restrict our study to a $\mathbb{Z}_2$ even sector featuring only three primaries $\{\mathbb{1}, \phi_{1,3}, \phi_{1,5}\}$ and their descendants. This QFT contains the minimum number of UV divergences that one could expect in a theory that requires coupling constant renormalization: the two-point function of $\phi_{1,5}$ renormalizes $\phi_{1,3}$ and the identity only; and there are no further UV divergences in integrated correlation functions with $\phi_{1,5}$ insertions. We are unaware of a simpler RG flow within the *unitary* minimal models, i.e. a unitary QFT that would only require second order coupling constant renormalization in a symmetry-selection sector with three or less primaries. Secondly, it has been conjectured that $M(3,7)$ deformed by $\phi_{1,5}$ flows to $M(3,5)$ [28] – a conjecture that we will support with a Hamiltonian Truncation calculation.

In Subsec. 6.1 we review the $M(3,7)$ theory, explain its Hamiltonian Truncation matrix and give the Effective Hamiltonian we use. In Subsec. 6.2 we demonstrate the convergence of the spectrum for a wide range of couplings. We also discuss in detail the various improve-

Table 3: The 6 primaries of the $M(3,7)$ minimal model along with their scaling dimensions and transformation properties under a global $\mathbb{Z}_2$ and $\mathcal{PT}$ symmetry. Here $X = +$ (i.e. even) or $X = -$ (i.e. odd) because the $\mathcal{PT}$ charges of the $\mathbb{Z}_2$ odd primaries is not unequivocally defined, but only their relative charges with respect to other $\mathbb{Z}_2$ odd primaries.

| $M(3,7)$ primaries | $\phi_{1,1}$ | $\phi_{1,2}$ | $\phi_{1,3}$ | $\phi_{1,4}$ | $\phi_{1,5}$ | $\phi_{1,6}$ |
|---|---|---|---|---|---|---|
| $\Delta$ | 0 | -5/14 | -2/7 | 3/14 | 8/7 | 5/2 |
| $\mathbb{Z}_2$ | Even | Odd | Even | Odd | Even | Odd |
| $\mathcal{PT}$ | Even | X | Odd | -X | Even | X |

ment terms added in the Effective Hamiltonian and their impact on convergence. Finally, in Subsec. 6.3 we use this Effective Hamiltonian to investigate the physics of the $M(3,7)$ flow. In particular we show that it has an intricate phase diagram and that at a boundary between the phases, the theory flows to the $M(3,5)$ CFT in the IR.

## 6.1 The Hamiltonian Truncation matrix

We summarize the CFT data which serves as the input of the HT calculation. The $M(3,7)$ model has central charge $c_{3,7} = -25/7$ and 6 distinct primary operators, which are labeled according to their position on the first row of the Kac table.

The fusion rules enjoy several discrete $\mathbb{Z}_2$ symmetries. The first is a global internal symmetry, which splits the Hilbert space of the CFT into an even and odd sector. The second is a realization of the discrete spacetime $\mathcal{PT}$ symmetry, which, in addition to acting on the primary fields, acts on $\mathbb{C}$-numbers as $i \to -i$.[11] The scaling dimensions, as well as the transformation properties of the primary fields under the global $\mathbb{Z}_2$ and $\mathcal{PT}$ are given in Tab. 3. The fusion rules of the fields in the $\mathbb{Z}_2$ even sector only are given by:

$$\phi_{1,3} \times \phi_{1,3} \sim \mathbb{1} + i\phi_{1,3} + \phi_{1,5} \qquad \phi_{1,3} \times \phi_{1,5} \sim \phi_{1,3} + i\phi_{1,5} \qquad \phi_{1,5} \times \phi_{1,5} \sim \mathbb{1} + i\phi_{1,3}, \quad (33)$$

where the imaginary OPE coefficients are indicated with a factor of $i$. Here we record the approximate values of the OPE coefficients between $\mathbb{Z}_2$-even primaries:

$$C_{333} \approx +1.97409\,i\,, \qquad C_{355} \approx +0.979627\,i\,, \qquad C_{335} \approx -0.113565\,. \quad (34)$$

The OPE coefficients between primaries in the even sector and all the fusion rules of the CFT are given in appendix C. The negative central charge, scaling dimensions and imaginary OPE coefficients are characteristics of the non unitarity of the $M(3,7)$ model. This implies that there are negative-normed states in the Hilbert space. Thus the Gram matrix of our basis states is not positive definite.

We will deform the $M(3,7)$ model by $\phi_{1,3}$ and $\phi_{1,5}$. These two relevant primaries are even. For the purposes of HT, for simplicity we will limit ourselves to study the even sector, which is composed of three primary families. The bare Hamiltonian is given by

$$H^{M(3,7)} + V = \frac{1}{R}\left(L_0 + \bar{L}_0 - \frac{c_{3,7}}{12}\right) + i\frac{g_3}{2\pi}\int_0^{2\pi R} dx\,\phi_{1,3}(0,x) + \frac{g_5}{2\pi}\int_0^{2\pi R} dx\,\phi_{1,5}(0,x). \quad (35)$$

---

[11]For completeness we provide a review of $\mathcal{PT}$-symmetry in appendix D.

Here $g_3, g_5$ are real dimensionful coupling constants with dimensions $2 - \Delta_3 = 16/7$ and $2 - \Delta_5 = 6/7$, respectively. Because $\phi_{1,3}$ is odd under $\mathcal{PT}$-symmetry, its coupling is purely imaginary, and therefore the full Hamiltonian is $\mathcal{PT}$-even.

Next we apply the renormalization procedure described in Sec. 2 at second order in the coupling $g_5$. We obtain the following contributions:

$$
\left( K_{\text{eff}, 2}^{M(3,7)} \right)_{fi} = \langle f | i \rangle \frac{g_5^2 R^{5/7}}{2} \Bigg[ \sum_{n=0}^{2n \leqslant \Delta'_{T,i}} + \sum_{n=0}^{2n \leqslant \Delta'_{T,f}} \Bigg] \frac{1}{2n + 8/7} \left( \frac{\Gamma(n + 8/7)}{\Gamma(n+1)\Gamma(8/7)} \right)^2 \tag{36}
$$

$$
+ \langle f | \phi_{1,3}(1) | i \rangle C_{55}^3 \frac{g_5^2 R^{5/7}}{2} \Bigg[ \sum_{n=0}^{2n \leqslant \Delta'_{T,i}} + \sum_{n=0}^{2n \leqslant \Delta'_{T,f}} \Bigg] \frac{1}{2n + 8/7} \left( \frac{\Gamma(n + 9/7)}{\Gamma(n+1)\Gamma(9/7)} \right)^2 .
$$

For this model, we define $\Delta'_{T,j} \equiv \Delta_T - 8/7 - \Delta_j$, as well as $\Delta''_{T,j} \equiv \Delta_T + 2/7 - \Delta_j$. We do not include $g_3^2$ and $g_3 g_5$ corrections because these are convergent, see Sec. 6.2 for further discussion. All in all the Hamiltonian Truncation matrix of the $\text{CFT}_{M(3,7)} + \int \phi_{1,5} + i \int \phi_{1,3}$ theory is given by

$$
H_{ij} = H_{ij}^{M(3,7)} + V_{ij} + \left( K_{\text{eff},2}^{M(3,7)} \right)_{ij} \quad \text{with} \quad \{\Delta_i, \Delta_j\} \leqslant \Delta_T . \tag{37}
$$

Besides including the needed counterterms at second order, we will also include the $\phi_{1,3}$ contribution to the Effective Hamiltonian at third order in $g_5$ in order to improve the convergence of the spectrum as a function of the cutoff. We note that $C_{55}^5 = 0$ and therefore the $\mathbb{1}$ contribution to $H_{\text{eff},3}$ vanishes. The third order result reads

$$
\left( K_{\text{eff},3}^{M(3,7)} \right)_{fi} = \langle f | \phi_{1,3}(1) | i \rangle C_{55}^3 C_{35}^3 \frac{g_5^3 R^{11/7}}{2} \Bigg[ \sum_{\substack{2K > \Delta'_{T,i} \\ 2Q > \Delta''_{T,i}}} \frac{B_{K,Q}^2}{X(K) Y(Q)} \tag{38}
$$

$$
- \sum_{\substack{K=0 \\ 2Q > \Delta''_{T,i}}}^{2K \leqslant \Delta'_{\Delta_T,i}} \frac{B_{K,Q}^2}{(Y(Q) - X(K)) Y(Q)} - \sum_{n=0}^{\infty} \frac{(r_n^{9/7})^2}{X(n)} \Bigg( \sum_{2n > \Delta'_{T,i}} \frac{(r_n^{4/7})^2}{X(n)} + \sum_{2n > \Delta''_{T,i}} \frac{(r_n^{4/7})^2}{Y(n)} \Bigg) \Bigg] + i \leftrightarrow f .
$$

We have defined $X(K) \equiv 2K - 8/7$ and $Y(Q) \equiv 2Q + 2/7$, and $r_n^\alpha \equiv \Gamma(n + \alpha)/(\Gamma(n+1)\Gamma(\alpha))$. Furthermore, $B_{K,Q} \equiv A_{K,Q} - A_{K-1,Q-1} - A_{K,Q-1} + A_{K-2,Q-2} - A_{K-1,Q-2} + A_{K-2,Q-2}$, with $A_{K,Q} \equiv \sum_{k=0}^{\text{Min}(K,Q)} r_k^{9/7} r_{Q-k}^{9/7} r_{K-k}^{9/7}$, and $A_{K,Q} \equiv 0$ if either $Q < 0$ or $K < 0$. Eq. (38) is derived in detail in appendix A.

Once the improvement terms up to third order in the coupling $g_5$ are included, the truncated Hamiltonian is given by

$$
H_{ij} + \left( K_{\text{eff},3}^{M(3,7)} \right)_{ij} , \tag{39}
$$

with $H$ given in (37). In (35), we have used the symbol $g_3$ to refer to the bare coupling, however in (37), (39) and in the rest of the section, we use the symbol to refer to the corresponding renormalised coupling.

## 6.2 Spectrum and cutoff dependence

In this section, we present the spectrum as a function of the truncation cutoff $\Delta_T$ for the $M(3,7) + i\phi_{1,3} + \phi_{1,5}$ theory at weak and strong coupling. We parametrize the two dimensional

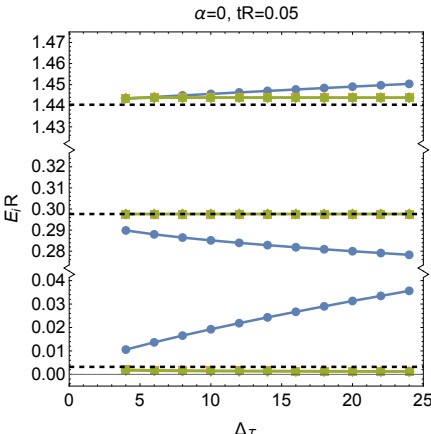

Figure 7: Lowest three energy levels for the $\mathcal{M}(3,7)$ minimal model deformed with its $\phi_{1,5}$ operator, at weak coupling $tR = 0.05$. The black dashed line denotes first–order perturbation theory, the blue refers to bare HT (35), the orange to HT with order $K_{\text{eff},2}$ (37) and the green to HT with $K_{\text{eff},2}$ and $K_{\text{eff},3}$ (39). At $\Delta_T = 24$, the truncated Hilbert space contains 7197 states.

space of couplings by $t$ (with units of energy) and a dimensionless angle $\alpha$, defined as

$$t\cos\alpha \equiv \text{sign}(g_5)|g_5|^{7/6}, \quad t\sin\alpha \equiv \text{sign}(g_3)|g_3|^{7/16}, \tag{40}$$

where here $g_3$ and $g_5$ refer to the renormalised couplings. The dimensionless parameter $tR = R\sqrt{\left(|g_5|^{7/6}\right)^2 + \left(|g_3|^{7/16}\right)^2}$ controls the coupling strength (and spatial volume) while the angle $\alpha$ parametrizes the relative proportion of both deformations. Taking $(tR, \alpha)$ as polar coordinates in a plane, RG flows are represented by curves from the origin to infinity, with monotonically increasing $tR$.

In Fig. 7 we plot the spectra that result from diagonalizing the bare Hamiltonian (blue), the renormalized (37) (orange) and the improved $H + K_{\text{eff},3}^{M(3,7)}$ in (38) (green). Horizontal dashed black lines show the result of first order perturbation theory. As expected, we find that the spectrum of the bare Hamiltonian diverges at large values of $\Delta_T$, while the renormalized (orange) and improved (green) converge to the perturbative result. The small gap between the numerical spectrum and perturbation theory is due to higher order perturbative corrections. [12] The difference between the renormalized (orange) and improved (green) spectrum lines is negligible because the coupling is weak. For illustration in Fig. 7 we only show a particular direction in the $\{g_3, g_5\}$ plane, i.e. $\alpha = 0$. We do find nice agreement with first order perturbation theory for all values of $\alpha$.

Having established a nice agreement with perturbation theory, we next inspect the cutoff dependence of the spectrum for larger values of the coupling $tR$. The result is shown in the panels of Fig. 8. In order to interpret these plots it is useful to take into account the large $\Delta_T$

---

[12]For the first and third state, $|\phi_{1,3}\rangle$ and $|\phi_{1,5}\rangle$ in terms of the UV-CFT labels, the next order correction is $O(g_5^3)$. For the identity state $|\mathbb{1}\rangle$, the first non-vanishing correction is $O(g_5^4)$. This observation explains why for these values of the couplings the HT prediction is indistinguishable from the unperturbed CFT.

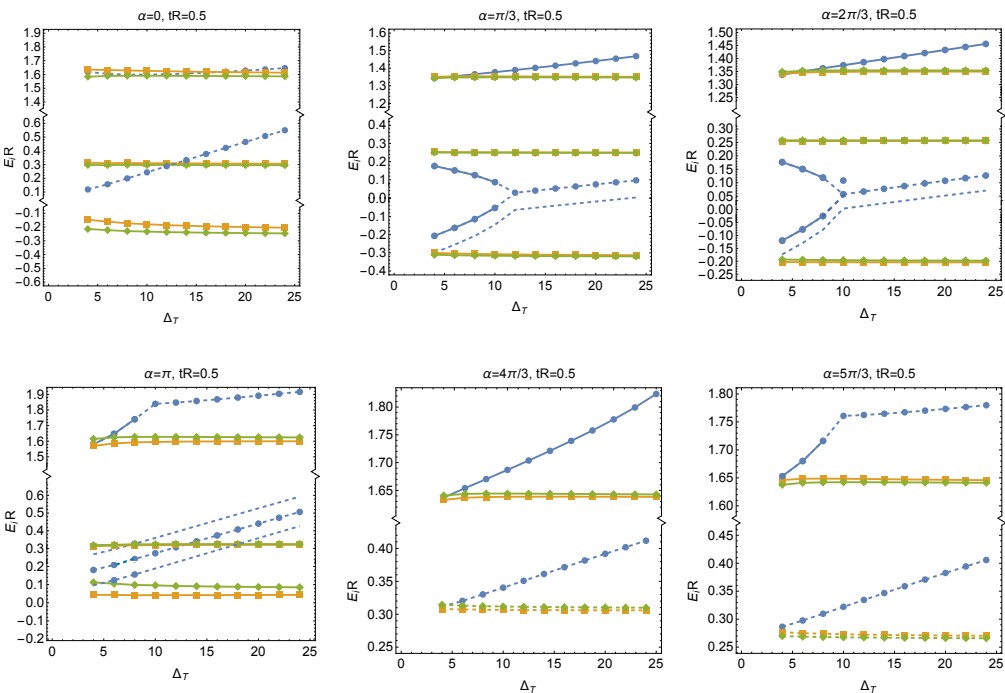

Figure 8: Spectrum at moderately strong coupling $tR = 0.5$ for various angles. Blue, orange and green refer to the calculations using the bare, order $g_5^2$, and order $g_5^3$ $K_{\text{eff}}$, respectively. Dashed lines denote complex eigenvalues, and we only show the real part.

asymptotics of the Effective Hamiltonian. The second order corrections are given by:

$$K_{\text{eff},2} \sim \underbrace{g_5^2 \left( \mathbb{1}\, \Delta_T^{2/7} + C_{55}^3\, \phi_{1,3}\, \Delta_T^{4/7} \right)}_{\text{included}} + \underbrace{g_3 g_5 \left( C_{35}^3\, \phi_{1,3}\, \Delta_T^{-6/7} + C_{55}^3\, \phi_{1,5}\, \Delta_T^{-16/7} \right)}_{\text{not included}}$$
$$+ \underbrace{g_3^2 \left( \mathbb{1}\, \Delta_T^{-18/7} + C_{33}^3\, \phi_{1,3}\, \Delta_T^{-16/7} + C_{33}^5\, \phi_{1,5}\, \Delta_T^{-26/7} \right)}_{\text{not included}} . \tag{41}$$

In this formula we only accounted for the local contribution from primary operators. The largest missing correction scales as $g_3 g_5 C_{35}^3 \Delta_T^{-6/7}$. However note that $|C_{35}^3| = O(1/10)$ while $|C_{35}^5| \approx |C_{33}^3| = O(1)$. Therefore corrections that appear subleading at large $\Delta_T$, such as $g_3 g_5 C_{55}^3 \Delta_T^{-16/7}$ and $g_3^2 C_{33}^3 \Delta_T^{-16/7}$, can have a similar impact on the spectrum for the values of $\Delta_T \sim O(10)$ considered in this work. Moving on, the third order corrections are given by:

$$K_{\text{eff},3} \sim \underbrace{g_5^3 \left( C_{55}^3 C_{35}^3 \Delta_T^{-2/7} \phi_{1,3} \right)}_{\text{included}} + \underbrace{g_5^3 \left( C_{55}^3 C_{35}^3 \Delta_T^{-12/7} \phi_{1,5} \right)}_{\text{not included}} + g_3\, O\!\left( \frac{g_5^2}{\Delta_T^{12/7}}, \frac{g_5 g_3}{\Delta_7^{22/7}}, \frac{g_3^2}{\Delta_7^{32/7}} \right) , \tag{42}$$

where the $\phi_{1,3}$ piece that we included is the leading contribution to $K_{\text{eff},3}$ at large $\Delta_T$. According to this power counting we expect that for $\alpha = 0, \pi$, (i.e. $g_3 = 0$), the effect of including $K_{\text{eff},3}$ should be more noticeable on the spectrum convergence. Indeed, in Fig. 8 we see that this expectation is realized.

More importantly, what we find is that for generic directions in the $\{g_3, g_5\}$ plane, including $K_{\text{eff}}$ leads to a nicely converged spectrum as $\Delta_T$ is increased. For all angles shown in Fig. 8

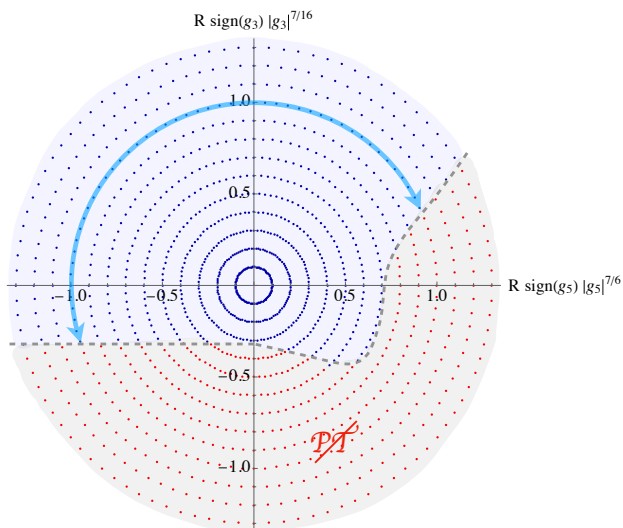

Figure 9: Phase diagram. Blue dots are values of the couplings for which the first four energy eigenvalues are real, and thus show no evidence of spontaneous breaking of $\mathcal{PT}$-symmetry. Red dots are values of the couplings for which at least one of the four lowest energy levels 'collides' into another to form a complex pair, signalling that $\mathcal{PT}$ symmetry is spontaneously broken.

the blue curves, corresponding to the bare Hamiltonian, show power-like divergences, while green and orange show convergence. We have found nice convergence of the spectrum when using $K_{\text{eff}}$ at (virtually) all angles and for couplings up to $tR \lesssim 1.5$.

We have plotted with dashed lines the real part of eigenvalues that acquire imaginary parts. We regard this effect as unphysical for the bare Hamiltonian (blue lines). However, for the converged spectrum (green and orange lines), this effect should be interpreted as the physical fingerprint of spontaneous $\mathcal{PT}$ breaking. Although we only show the real part of the complex spectra, their imaginary parts are equally well converged.

Having demonstrated the capability of the $K_{\text{eff}}$ formalism to compute the spectrum of this QFT at strong coupling, we next turn to discuss some of the most salient physics of the $M(3,7) + i\phi_{1,3} + \phi_{1,5}$ flow.

## 6.3 Phase diagram and EFT interpretation

In this section we study the phase diagram of $M(3,7) + i\phi_{1,3} + \phi_{1,5}$ as a function of the couplings $\{g_3, g_5\}$, by mapping out the values of the couplings for which the theory presents spontaneous breaking of $\mathcal{PT}$-symmetry. The $\mathcal{PT}$-symmetry is realized by an anti-unitary operator. Therefore, the presence of $\mathcal{PT}$-symmetry, $[\mathcal{PT}, H] = 0$, does not guarantee that the eigenvalues of the Hamiltonian are eigenvalues of $\mathcal{PT}$. It is however a consequence of the symmetry that the eigenvalues of the Hamiltonian are either real or come in complex conjugate pairs.

The $\mathcal{PT}$-symmetry preserving phase is characterized by a real spectrum of the Hamiltonian, which in turn implies that the Hamiltonian eigenstates are also eigenstates of $\mathcal{PT}$. Instead the $\mathcal{PT}$-breaking phase is characterized by the appearance of complex conjugate pairs of eigenvalues, or equivalently, by Hamiltonian eigenstates $|E\rangle$ that are are related by $\mathcal{PT}|E\rangle = |E^*\rangle$. See appendix D for a review.

Table 4: Summary of $M(3,5)$ minimal model operator content.

| M(3,5) primaries | $\phi_{1,1}$ | $\phi_{1,2}$ | $\phi_{1,3}$ | $\phi_{1,4}$ |
|---|---|---|---|---|
| $\Delta$ | 0 | -1/10 | 2/5 | 3/2 |
| $\mathbb{Z}_2$ | Even | Odd | Even | Odd |
| $\mathcal{PT}$ | Even | X | Odd | -X |

In Fig. 9 we show the phases of $\mathcal{PT}$-symmetry as a function of the couplings. We computed the four lowest energy eigenstates in the $\mathbb{Z}_2$-even sector. [13] The red dots correspond to the phase where $\mathcal{PT}$-symmetry is spontaneously broken because at least one of the lowest four eigenvalues is complex. Instead the blue dots are values of the couplings for which the four lowest eigenstates in the $\mathbb{Z}_2$-even sector are real and therefore do not show evidence of $\mathcal{PT}$-symmetry breaking. The shaded grey and blue regions, and the dashed grey curve are there in order to guide the eye.

The phase transition boundary can be close to the dashed gray line of Fig. 9, but must be inside the blue region. This is because the criterion for $\mathcal{PT}$-breaking that we use considers only the four lowest energy states and not the whole spectrum. On the left boundary it is the ground state that becomes complex, while on the right boundary only excited states become complex. At the left endpoint of the boundary between the phases (i.e. in the deep IR), we may expect the IR QFT to be described by a CFT. At this boundary, the ground state merges with the first excited state energy, which is consistent with what we would expect if the theory at this boundary is an ungapped CFT. On the right boundary, the first gap remains nonvanishing even for large radius, disfavoring a CFT interpretation. Evidence for critical and non-critical $\mathcal{PT}$-breaking-transitions have been recently studied using TCSA [29, 30].

Next we will determine which IR CFTs can be reached from the $M(3,7)$ CFT. We first employ a generalization of the $c$-theorem (originally derived for unitary flows) to $\mathcal{PT}$-symmetric RG flows. One can define a monotonically decreasing function along a non-unitary but $\mathcal{PT}$ invariant RG flow, which at the fixed points equals the effective central charge of the CFT, defined by $c_{\text{eff}} \equiv c - 12\Delta_{\min}$ for rational CFTs [31]. The effective central charge is positive and equals $c_{\text{eff}} = 1 - \frac{6}{pq}$ for the minimal model $M(p,q)$. For our flow, the UV CFT has $p \cdot q = 3 \cdot 7 = 21$, hence candidate minimal models for the IR CFT must have $pq < 21$. Excluding unitary minimal models, this criterium reduces our search to four non-trivial IR CFTs

$$M(2,5), \ M(2,7), \ M(2,9) \text{ and } M(3,5) \,. \tag{43}$$

Finally note that while $M(3,7)$ has a global $\mathbb{Z}_2$ symmetry, none of the models $M(2,5)$, $M(2,7)$ or $M(2,9)$ do. Therefore the only possible CFT that $M(3,7) + i\phi_{13} + \phi_{15}$ can flow to in the IR is the $M(3,5)$. Next we will show that this expectation is nicely realized by our data, providing a non-trivial example of Hamiltonian Truncation theory for UV divergent QFTs.

*6.3.1  $M(3,5)$ Effective Field Theory*

The $M(3,5)$ minimal model has central charge $c_{3,5} = -3/5$ and contains 4 independent primaries, denoted by $\phi_{1,i}, i = 1, \ldots, 4$. Their scaling dimensions and their transformation properties under a global $\mathbb{Z}_2$ and $\mathcal{PT}$ symmetry is given in Tab. 4. For this work we will only

---

[13]The data is taken with $\Delta_T = 18$ using (39), and is well converged in the cutoff $\Delta_T$.

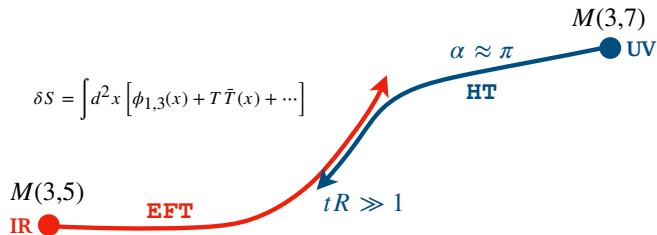

Figure 10: Schematic representation of the calculation presented in this section: we have computed the spectrum of the $M(3,7) + ig_3\phi_{1,3} + g_5\phi_{1,5}$ theory at strong coupling using Hamiltonian Truncation (HT). Then we interpret the HT data in the IR, and in the direction $\alpha = \pi$ (defined in (40)), in terms of an Effective Field Theory (EFT) defined by perturbing the $M(3,5)$ CFT by its single $\mathbb{Z}_2$-even relevant operator and the leading $\mathbb{Z}_2$-even irrelevant operator.

focus on the even sector of the CFT, for which the only non trivial fusion rule is $\phi_{1,3} \times \phi_{1,3} \sim \mathbb{1}$. The full fusion rules are given in appendix C for completeness.

Since the two deformations of the UV CFT are $\mathbb{Z}_2$ even, we conclude that in the IR $M(3,5)$ EFT we can only include fields in the $\mathbb{1}$ and $\phi_{1,3}$ family. We thus expect the relevant operator $\phi_{1,3}$ of dimension 2/5 to control off-criticality in the EFT fit. The leading irrelevant operator is $L_{-2}\bar{L}_{-2}\mathbb{1}$. The next one is the second level descendant of $\phi_{1,3}$, $L_{-2}\bar{L}_{-2}\phi_{1,3}$ of dimension $4 + 2/5$. In order to preserve $\mathcal{PT}$ symmetry, the couplings of operators in the $\phi_{1,3}$ family have to be purely imaginary. Moreover, in the IR conformal perturbation theory, the first order correction of any operator in the $\phi_{1,3}$ conformal family to any of the gaps in the even sector vanishes.

All in all, the EFT action reads:

$$S_{M(3,5)} + \int d^2x \left[ \Lambda^2 + i\frac{g_\phi}{2\pi}M^{8/5}\phi_{1,3}(x) + \frac{g_{T\bar{T}}}{2\pi M^2}T\bar{T}(x) + \mathcal{O}(M^{-12/5}) \right]. \tag{44}$$

We remark that even though the next irrelevant operator is a second descendant of $\phi_{1,3}$, as we just explained, it does not affect the even gaps at tree level because of selection rules. The next irrelevant operator that affects the even gaps at tree level is the level four, dimension eight descendant of the identity. These lucky coincidences make the spectrum of EFT operators quite sparse, and therefore the Effective Action in (44) should be very effective in characterizing the last steps of the RG flow.

Next we set $M = 1$ and compute the first three $\mathbb{Z}_2$-even gaps, using the EFT action (44). We need the corrections to the energy levels of the states $|\mathbb{1}\rangle, |\phi_{1,3}\rangle, |L_{-1}\bar{L}_{-1}\phi_{1,3}\rangle$ and $|L_{-2}\bar{L}_{-2}\mathbb{1}\rangle$. These states are orthogonal to any other state in the $M(3,5)$ CFT and are also non-degenerate. Therefore we can use standard Rayleigh-Schrödinger perturbation theory. [14] The first three gaps are then given by

$$\delta E_i^{\text{EFT}} \equiv (E_i^{\text{EFT}} - E_0^{\text{EFT}})R = \Delta_i + (ig_\phi)^2(\beta_i - \beta_0) + g_{T\bar{T}}\left[(E_iR/2)^2 - (E_0R/2)^2\right]. \tag{45}$$

For our current purposes it is enough to compute the $\beta_i$'s numerically by summing over states as instructed by Rayleigh-Schrödinger perturbation theory. Because the perturbation is very

---

[14]The formula we used reads: $E_i^{(2)} = \sum_{m,n\neq i} \frac{\langle i|V|m\rangle(G^{-1})_{mn}\langle n|V|i\rangle}{\langle i|i\rangle(E_i^{(0)} - E_n^{(0)})}$, with $G_{ij} = \langle i|j\rangle$. The inverse Gram matrix accounts for the non orthonormality of the states $|m \neq i\rangle$, and we have also divided by the norm of the state $|i\rangle$.

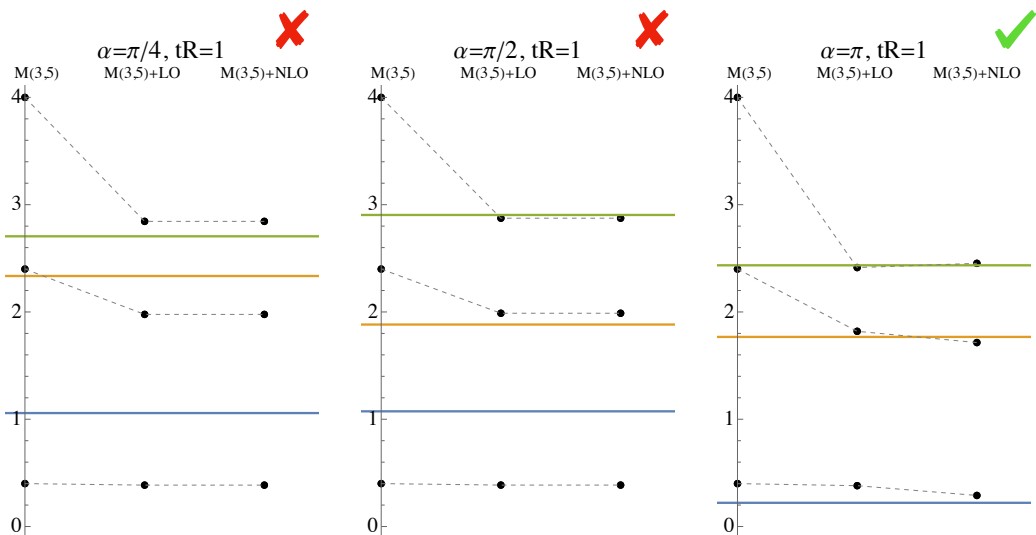

Figure 11: EFT fits to the first three gaps in the $M(3,7) + i\phi_{1,3} + \phi_{1,5}$ QFT at $tR = 1$, for three values of $\alpha$. The three fits are done over the first three gaps, with the two parameters $g_{T\bar{T}}$ and $g_\phi$. We observe that for $\alpha = \pi$ the data is well described by the $M(3,5)$ EFT. For comparison we included two other directions of the flow $\alpha = \pi/4$, $\alpha = \pi/2$, which are not well described by the $M(3,5)$ EFT.

relevant ($\Delta_{\phi_{1,3}} = 2/5$) the convergence of such sums is very fast. We obtain

$$\beta_{\mathbb{1}} = -2.60, \quad \beta_{\phi_{1,3}} = 2.48, \quad \beta_{L_{-1}\bar{L}_{-1}\phi_{1,3}} = -0.41, \quad \beta_{T\bar{T}} = -15.30. \tag{46}$$

Next we use (45) to fit the Hamiltonian Truncation data. For clarity in Fig. 10 we include a diagram summarising the logic of the exercise presented in this section. After having explained that the HT data has negligible cutoff corrections (blue arrow) for $tR \lesssim 1$, we now focus on the phenomenology. Thus we attempt to interpret the IR data in terms of field theoretic computations (red arrow).

In Fig. 11 we show the result of the fit obtained by minimizing the function $\chi^2(g_\phi, g_{T\bar{T}}) = \sum_{i=1}^{3} [\delta E_i^{\mathrm{HT}} - \delta E_i^{\mathrm{EFT}}(g_\phi, g_{\bar{T}T})]^2$ for three values of the angle $\alpha$ in the plane of Fig 9. The three angles lie in the blue region of Fig 9. The horizontal lines of Fig. 11 are the gaps obtained with HT (using (39)). The left-most column of black dots are the $M(3,5)$ gaps, the central column of black dots are obtained by doing the fit with the tree-level formula (i.e. with $g_\phi = 0$), while the rightmost column of black dots is the NLO fit with two couplings, i.e. using the function (45). The fit is non-trivial as it involves one or two free parameters and three different states.

We note that the NLO fit for $\alpha = \pi$ is much better than for the other angles shown in Fig. 11. This has a clear explanation following from the structure of perturbation theory. Note that the $T\bar{T}$ correction is 'universal' and pushes all the spectrum in the same direction (either upwards or downwards, depending on the sign of $g_{T\bar{T}}$). This fact alone explains why the LO fit of $\alpha = \pi$ is much better than the LO fits at $\alpha = \pi/4$ and $\alpha = \pi/2$. Furthermore, the NLO parameters $\beta_i$'s have the right structure to give the final good kick to the energy levels for $\alpha = \pi$. This is because $\beta_{\phi_{1,3}} - \beta_0 > 0$ and $\beta_{L_{-1}\bar{L}_{-1}\phi_{1,3}} - \beta_0 > 0$ are positive while $\beta_{T\bar{T}} - \beta_0 < 0$ is negative. Thus the second order correction pushes the first two gaps down while the third gap is pushed upwards. This is the wrong correction to the spectrum if the EFT had to explain the deviations of the energy $\delta E_i^{\mathrm{EFT}} - \delta E_i^{HT}$ levels at $\alpha = \pi/4, \pi/2$, while this correction is

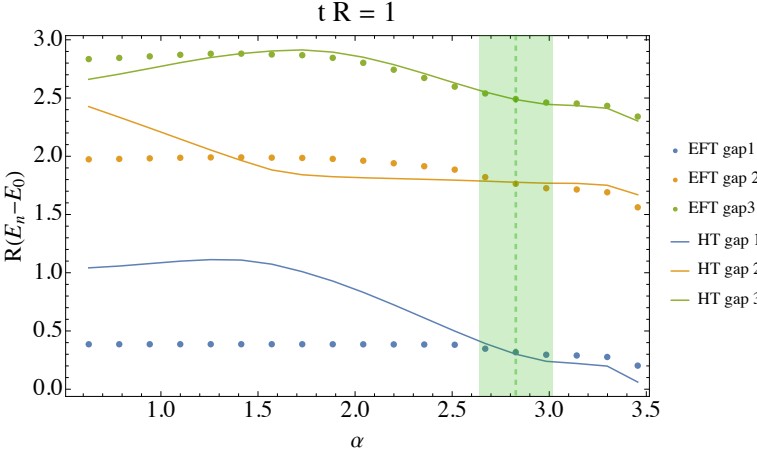

Figure 12: Comparison of HT data (solid lines) with NLO fits (dots) as a function of $\alpha$.

just right for the spectrum $\alpha = \pi$. For reference, we get $|g_\phi| = 0.13$ and $g_{T\bar{T}} = -0.43$, which according to (45) indicates that the EFT corrections to the gaps are perturbative. The sizes of the EFT corrections can also be seen in the right panel of Fig. 11.

We have not investigated the radius dependence of the spectrum in this work. For $\alpha$ tuned to the critical value, the energy gaps should converge at large radius to the scaling dimensions of the $M(3,5)$ CFT divided by the radius. In the future it would be interesting to study this scaling with radius, along with that of the leading EFT corrections.

We point out that Ref. [28] provided a non-linear integrable equation that interpolates the often-called finite-volume effective central charge (i.e. cylinder finite-volume vacuum energy) of $M(3,7)$ and $M(3,5)$ along the $M(3,7) + \phi_{1,5}$ direction. Our calculation can be viewed as a field-theoretic calculation, in a particular HT scheme, that supports the conjecture implied by that result.

From the fits just presented we conclude that in the IR, and close to the boundary between the $\mathcal{PT}$-symmetrric and $\mathcal{PT}$-breaking phases at $\alpha \approx \pi$, the low energy spectrum of $M(3,7) + i\phi_{1,3} + \phi_{1,5}$ is well described by the $M(3,5)$ EFT. For $\alpha$ tuned precisely to this boundary between phases, the QFT will flow to the $M(3,5)$ CFT in the infrared (with the coupling of the single relevant deformation in the $M(3,5)$ EFT tuned to zero).

Further evidence for this point is provided in Fig. 12. The curves shown in the plot are the first three gaps obtained with HT, as a function of $\alpha$. The angle $\alpha$ is varied along the section indicated by a blue arrow in Fig. 9. At discrete values of $\alpha$ we show a column of three dots that correspond to the best fit NLO values (i.e. analogous to the rightmost column of Fig. 11). From this plot we infer that, indeed, the $\mathcal{PT}$-breaking-boundary at around $\alpha \approx \pi/4$ does not show critical behavior, while the other boundary at $\alpha \approx \pi$ is well described by the $M(3,5)$ EFT!

## 7  Conclusions & outlook

We used Hamiltonian Truncation to study three example QFTs at strong coupling in 1+1 dimensions. Each QFT had UV divergences requiring renormalization, so we employed the formalism

developed in Ref. [12] to systematically derive the necessary (nonlocal) counterterms to obtain the renormalized Effective Hamiltonian for each QFT. In this way, we approximated the infinite dimensional Hamiltonian matrices of the original QFTs with finite-dimensional (truncated) matrices, where every element was free of UV divergences.

Effective Hamiltonians act on a truncated Hilbert space spanned by states of a solvable QFT with energies below a given cutoff scale. We verified with explicit numerical calculations that the low eigenvalues of our Effective Hamiltonians converged to definite values as this cutoff was increased, indicating that our formalism allows for consistent determination of the low-energy spectrum of each QFT.

First, we looked at the Ising CFT deformed with its $\epsilon$ operator. We then used Effective Hamiltonians to compute both the low-energy spectrum and Zamolodchikov $c$-function, checking they were consistent with the known results. In particular, using the Effective Hamiltonian gave a more accurate determination for the $c$-function for a fixed, finite cutoff than using the simpler bare Hamiltonian.

Then we studied the Tricritical Ising CFT deformed by its $\epsilon'$ primary operator. Under the renormalization group, this QFT flows to the Ising CFT in the low-energy limit. Using Effective Hamiltonians, we calculated the low energy spectrum and checked that it was well described by an Ising Effective Field Theory (EFT) – a QFT constructed as the Ising CFT deformed with irrelevant operators. We also computed the Zamolodchikov $c$-function, finding that adding an improvement term that vanishes in the infinite cutoff limit improved the accuracy of our calculations at finite cutoff further.

Finally, we investigated the $M(3,7)$ minimal model CFT deformed with two relevant operators. This QFT features renormalization of a relevant coupling besides the cosmological constant unlike the other two QFTs, and so provides an extra consistency check on the Effective Hamiltonian formalism. The QFT is also nonunitary and can enter a phase in which $\mathcal{PT}$ symmetry is spontaneously broken in part of its parameter space. After verifying that we could obtain a consistent low-energy spectrum using an Effective Hamiltonian, we studied the phase diagram. We then provided evidence that this QFT flows to the $M(3,5)$ CFT under the renormalization group, after tuning couplings to one of the phase boundaries.

The systematic procedure we demonstrate for calculating counterterms clears a significant obstacle restricting investigation of other QFTs with UV divergences. The use of renormalized Effective Hamiltonians then opens numerous potential research avenues – both in the arena of 2d minimal models and higher dimensional CFTs with relevant deformations. For example, $\phi^4$ in $d = 2+1$ dimensions requires mass renormalization and it would be interesting to revisit the studies in Refs. [32,33] using this framework. We look forward to finding out much more about strongly-coupled QFTs in the future.

## Acknowledgements

We are grateful to A. Liam Fitzpatrick and Matthew T. Walters for useful discussions. We thank Slava Rychkov and Bernardo Zan for useful comments on the draft. This project has received support from the European Research Council, grant agreement n. 101039756.

## A More details in $K_{\text{eff}}$ computation

In this appendix we provide a detailed derivation of the various $K_{\text{eff}}$ operators we use throughout this work. We consider only deformations by a primary operator $V = \frac{g}{2\pi} \int_0^{2\pi R} dx \, \phi_\Delta(0, x)$ satisfying $4/3 > \Delta \geqslant 1$, for which the only UV divergences are at second order in $g$.

### A.1 Renormalization with local regulator

We begin by specifying our local renormalization scheme as advocated in step one of Sec. 2. We regulate correlation functions of the deforming operator $\phi_\Delta$ with a short distance cutoff $\epsilon > 0$: $\langle \phi_\Delta(x_1) \dots \phi_\Delta(x_n) \rangle \to \langle \phi_\Delta(x_1) \dots \phi_\Delta(x_n) \rangle \prod_{i<j}^n \theta(|x_i - x_j| - \epsilon)$. Then, we add to the Hamiltonian a set of local regulator-dependent counterterms in order to obtain a finite spectrum as $\epsilon \to 0$. The renormalized Hamiltonian reads:

$$H(\epsilon) = H_{\text{CFT}} + \frac{g}{2\pi} \int_0^{2\pi R} dx \, \phi_\Delta(0, x) + \sum_{\substack{\mathcal{O}_i \in \phi_\Delta \times \phi_\Delta \\ 2\Delta - 2 - \Delta_{\mathcal{O}_i} \geqslant 0}} \left( \lambda_{\text{ren.}}^{\mathcal{O}_i} + \left( \frac{g}{2\pi} \right)^2 \lambda_{\text{c.t.}}^{\mathcal{O}_i}(\epsilon) \right) \int_0^{2\pi R} dx \, \mathcal{O}_i(0, x).$$

(A.1)

We choose our counterterms to be:

$$\lambda_{\text{c.t.}}^{\mathcal{O}_i}(\epsilon) \equiv R^{2+\Delta_{\mathcal{O}_i}-2\Delta} \int_{\substack{0 \leqslant |x| \leqslant 1 \\ |1-x| > \epsilon}} d^2 x |x|^{\Delta-2} \langle \mathcal{O}_i(\infty) \phi_\Delta(1) \phi_\Delta(x) \rangle.$$

(A.2)

Whenever $\mathcal{O}_i = \mathbb{1}$, we set $\lambda_{\text{ren.}}^{\mathbb{1}} = 0$. In the RG flows we consider, the only non trivial operator requiring renormalization is the $\phi_{1,3}$ in the $M(3, 7)$ flow, and in this case $\lambda_{\text{ren.}}^{\phi_{1,3}} \equiv i g_3/(2\pi)$. The three point function of primaries can be expanded in powers of $|x|$ as follows:

$$\langle \mathcal{O}_i(\infty) \phi_\Delta(1) \phi_\Delta(x) \rangle = \frac{C_{\Delta\Delta}^{\mathcal{O}_i}}{|1-x|^{2\Delta-\Delta_{\mathcal{O}_i}}} = C_{\Delta\Delta}^{\mathcal{O}_i} \sum_{n=0}^{\infty} |x|^n C_n^{\Delta-\Delta_{\mathcal{O}_i}/2}(\cos\theta).$$

(A.3)

The $C_n^\alpha$ are the Gegenbauer polynomials, and $\theta$ is the angle between the two dimensional vector $x$ and some arbitrary unit vector in the plane denoted by 1. Now we can write the counterterm coupling in terms of an infinite sum:

$$\lambda_{\text{c.t.}}^{\mathcal{O}_i}(\epsilon) = 2\pi R^{2+\Delta_{\mathcal{O}_i}-2\Delta} C_{\Delta\Delta}^{\mathcal{O}_i} \sum_{n=0}^{\infty} \frac{u_n^{\Delta-\Delta_{\mathcal{O}_i}/2,\epsilon}}{2n+\Delta},$$

(A.4)

where we have defined the following integrals:[15]

$$u_n^{\alpha,\epsilon} \equiv \frac{(2n+\Delta)}{2\pi} \int_{\substack{0 \leqslant |x| \leqslant 1 \\ |1-x| > \epsilon}} d^2 x |x|^{2n+\Delta-2} C_{2n}^\alpha(\cos\theta), \quad u_n^\alpha \equiv \lim_{\epsilon \to 0} u_n^{\alpha,\epsilon}.$$

(A.5)

It can be checked that the series in A.4 diverges in the $\epsilon \to 0$ limit if and only if $2\Delta - 2 - \Delta_{\mathcal{O}_i} \geqslant 0$ (see [14] for more details).

---

[15]Note a slight change in definition with respect to the same variable in [12]: $(u_n^{\alpha,\epsilon})_{\text{here}} = \frac{1}{4\pi}(u_n^{\alpha,\epsilon})_{\text{there}}$ in the case $d = 2$.

## A.2  $K_{\text{eff},2}$

Having fixed the renormalized Hamiltonian $H(\epsilon)$, we now compute the second order contribution to the Effective Hamiltonian introduced in Eq. (4):

$$(\Delta H_2(\epsilon))_{ij} = \frac{R}{2} V(\epsilon)_{ih} \left( \frac{1}{\Delta_{ih}} + \frac{1}{\Delta_{jh}} \right) V(\epsilon)_{hj}, \tag{A.6}$$

taking $V(\epsilon) = H(\epsilon) - H_{\text{CFT}}$ from (A.1) and keeping *only* terms that are $\mathcal{O}(g^2)$.

While the $\sum_{n\geqslant 2} \Delta H_n(\epsilon)$ sum is defined as an expansion in powers of $V(\epsilon)$, $\sum_{n\geqslant 2} H_{\text{eff},n}(\epsilon)$ is defined as an expansion in powers of $g$. The two differ in our case because schematically, $V(\epsilon) \sim g\phi_\Delta + g^2\lambda^{\mathcal{O}_i}\mathcal{O}_i$. $H_{\text{eff},n}(\epsilon)$ will receive contributions from all $\Delta H_m(\epsilon)$ for $\lceil n/2 \rceil \leqslant m \leqslant n$. For $n = 2$ only $\Delta H_2(\epsilon)$ contributes, and neglecting terms of higher order than $\mathcal{O}(g^2)$ amounts to computing only the $\phi_\Delta \phi_\Delta$ contribution from $\Delta H_2(\epsilon)$.

The only term in $\Delta H_2(\epsilon)$ at order $g^2$ is (modulo a trivial symmetrization of the $fi$ indices):

$$R \sum_{\Delta_h > \Delta_T} \frac{V(\epsilon)_{fh} V(\epsilon)_{hi}}{\Delta_{ih}} \bigg|_{\mathcal{O}(g^2)} = -\frac{g^2}{2\pi} \sum_{\Delta_h > \Delta_T} \int_{\substack{\tau < 0,\epsilon \\ 0 \leqslant x \leqslant 2\pi R}} d\tau dx \, e^{\tau \frac{\Delta_h - \Delta_i}{R}} \langle f | \phi_\Delta(0, \kappa) | h \rangle \langle h | \phi_\Delta(0, x) | i \rangle. \tag{A.7}$$

Here $\kappa$ is an arbitrary point on the circle, $\kappa \in [0, 2\pi R]$ and the $\epsilon$ subscript means that we are regulating all integrated correlators with the short distance cutoff $\epsilon$.

Using $\phi_\Delta(\tau, x) = e^{tH_0}\phi_\Delta(0, x)e^{-tH_0}$ and Weyl transforming the integral from the cylinder to the plane,[16] we obtain

$$\frac{R}{2} \sum_{\Delta_h > \Delta_T} \frac{V_{fh} V_{hi}}{\Delta_{ih}} \bigg|_{\epsilon, \mathcal{O}(g^2)} = -\frac{g^2}{4\pi} R^{3-2\Delta} \int_{\substack{0 \leqslant |x| \leqslant 1 \\ |1-x| > \epsilon}} d^2x |x|^{\Delta-2} \langle f | \phi_\Delta(1) \stackrel{!}{\cdot} \phi_\Delta(x) | i \rangle. \tag{A.8}$$

We denote the partial resolution of the identity $\stackrel{!}{\cdot} \equiv \sum_{\Delta_h > \Delta_T} |h\rangle \langle h|$. At this point our computation has been exact. We will now perform a local approximation of the $H_{\text{eff},2}$ operator. We start by noting that, in the region $|1-x| < 1$, the correlator in (A.8) can be computed using the OPE of the two $\phi_\Delta$ fields:[17]

$$\langle f | \phi_\Delta(1) \stackrel{!}{\cdot} \phi_\Delta(x) | i \rangle = \sum_{\mathcal{O}} \langle f | \mathcal{O}(1) | i \rangle \langle \mathcal{O}(\infty)\phi_\Delta(1) \stackrel{!}{\cdot}^i \phi_\Delta(x) \rangle, \qquad |1-x| < 1. \tag{A.9}$$

The partial insertion of the identity is computed by expanding the full correlator in powers of $|x|$ and keeping only terms with powers of $|x|$ greater than $\Delta'_{T,i} \equiv \Delta_T - \Delta - \Delta_i$, hence the $i$ superscript in the partial insertion. We will now make the approximation of replacing the integrand in (A.8) by (A.9) over the *entire* domain of integration of (A.8), keeping only a finite amount of contributions from local operators $\mathcal{O}$.

We have performed three approximations: the first is extending (A.9) to the region $|1-x| > 1$, the second is discarding the integral of the original integrand over the region $|1-x| > 1, |x| \leqslant 1$, and the third is truncating the sum over operators in (A.9). The first two of these changes are manifestly $\epsilon$ independent, thus do not spoil the cancellation of UV divergences.[18] Regarding

---

[16] The metrics in the plane and the cylinder are related by $ds^2_{\text{cylinder}} = \left(\frac{R}{r}\right)^2 ds^2_{\text{plane}}$.

[17] This formula, among with the generalization necessary for computing the local approximation to the Effective Hamiltonian at higher orders in the coupling, is derived in [12].

[18] The error from the first two approximations should be exponentially suppressed in the cutoff $\Delta_T$ since only arbitrarily high energy states $|h\rangle \langle h|$ are inserted between fields $\phi_\Delta(1)$ and $\phi_\Delta(x)$, which are a finite distance away for $|1-x| > 1$.

the truncation of operators in the OPE, it is of course crucial to include all those satisfying $2\Delta - 2 - \Delta_{\mathcal{O}} \geqslant 0$, for which the integral $\int d^2x \langle \mathcal{O}(\infty)\phi_\Delta(1)\phi_\Delta(x)\rangle$ diverges as $\epsilon \to 0$. The rest of the operators' contributions are finite as $\epsilon \to 0$, vanish as $\Delta_T \to \infty$ with a power law, and thus can be seen as improvement terms that speed up convergence in $\Delta_T$.

All in all, integrating (A.9) over the whole integration region using the expansion (A.3) and using the definition of $u_n^{\alpha,\epsilon}$ given in (A.5), we can write

$$\frac{R}{2} \sum_{\Delta_h > \Delta_T} \frac{V_{fh} V_{hi}}{\Delta_{ik}} \bigg|_{\epsilon, \mathcal{O}(g^2)} = -\frac{g^2}{2} R^{3-2\Delta} \sum_{\mathcal{O}_j} C_{\Delta\Delta}^{\mathcal{O}_j} \langle f | \mathcal{O}_j(1)|i\rangle \sum_{2n > \Delta'_{T,i}} \frac{u_n^{\Delta - \Delta_{\mathcal{O}_j}/2, \epsilon}}{2n + \Delta},$$

up to $\epsilon$ independent terms exponentially supressed in $\Delta_T$. All in all we have

$$(\Delta H_2(\epsilon))_{fi}\bigg|_{\mathcal{O}(g^2)} = -g^2 R^{3-2\Delta} \sum_{\mathcal{O}_j} \frac{C_{\Delta\Delta}^{\mathcal{O}_j}}{2} \langle f | \mathcal{O}_j(1)|i\rangle \left( \sum_{2n > \Delta'_{T,i}} + \sum_{2n > \Delta'_{T,f}} \right) \frac{u_n^{\Delta - \Delta_{\mathcal{O}_j}/2, \epsilon}}{2n + \Delta}. \quad \text{(A.10)}$$

The sum over operators $\mathcal{O}_j$ runs over any primaries appearing in the $\phi_\Delta \times \phi_\Delta$ OPE that we chose to include in our local approximation of $\Delta H_2(\epsilon)$. At this point we add the $\mathcal{O}(g^2)$ local counterterm(s) present in $V(\epsilon)$ and we obtain

$$\left( H_{\text{eff},2}(\epsilon) \right)_{fi} = g^2 R^{3-2\Delta} \Bigg[ \sum_{\substack{\mathcal{O}_j \in \phi_\Delta \times \phi_\Delta \\ 2\Delta - 2 - \Delta_{\mathcal{O}_j} \geqslant 0}} \frac{C_{\Delta\Delta}^{\mathcal{O}_j}}{2} \langle f | \mathcal{O}_j(1)|i\rangle \left( \sum_{2n \leqslant \Delta'_{T,i}} + \sum_{2n \leqslant \Delta'_{T,f}} \right) \frac{u_n^{\Delta - \Delta_{\mathcal{O}_j}/2, \epsilon}}{2n + \Delta}$$

$$- \sum_{\substack{\mathcal{O}_j \in \phi_\Delta \times \phi_\Delta \\ 2\Delta - 2 - \Delta_{\mathcal{O}_j} < 0}} \frac{C_{\Delta\Delta}^{\mathcal{O}_j}}{2} \langle f | \mathcal{O}_j(1)|i\rangle \left( \sum_{2n > \Delta'_{T,i}} + \sum_{2n > \Delta'_{T,f}} \right) \frac{u_n^{\Delta - \Delta_{\mathcal{O}_j}/2, \epsilon}}{2n + \Delta} \Bigg],$$

$$\text{(A.11)}$$

which contains all the $\mathcal{O}(g^2)$ pieces. The first line contains a finite sum from UV divergent operators, while the second line contains UV finite improvement terms.

We can now finally take the $\epsilon \to 0$ limit in (A.11) analytically to obtain the $K_{\text{eff},2} \equiv \lim_{\epsilon \to 0} H_{\text{eff},2}(\epsilon)$ operator:

$$\left( K_{\text{eff},2} \right)_{fi} = g^2 R^{3-2\Delta} \Bigg[ \sum_{\mathcal{O}_j \in \{\text{div.}\}} C_{\Delta\Delta}^{\mathcal{O}_j} \langle f | \mathcal{O}_j(1)|i\rangle \frac{P(\Delta'_{T,i}, \bar{\Delta}_j) + P(\Delta'_{T,f}, \bar{\Delta}_j)}{2}$$

$$- \sum_{\mathcal{O}_j \in \{\text{conv.}\}} C_{\Delta\Delta}^{\mathcal{O}_j} \langle f | \mathcal{O}_j(1)|i\rangle \frac{2S(\bar{\Delta}_j) - P(\Delta'_{T,i}, \bar{\Delta}_j) - P(\Delta'_{T,f}, \bar{\Delta}_j)}{2} \Bigg]. \quad \text{(A.12)}$$

We have defined $\bar{\Delta}_j \equiv \Delta - \Delta_{\mathcal{O}_j}/2$, and the partial sums

$$P(\Delta_T, \bar{\Delta}) \equiv \sum_{2n \leqslant \Delta_T} \frac{u_n^{\bar{\Delta}}}{2n + \Delta} = \sum_{2n \leqslant \Delta_T} \left( \frac{\Gamma(\bar{\Delta} + n)}{\Gamma(n+1)\Gamma(\bar{\Delta})} \right)^2 \frac{1}{2n + \Delta}. \quad \text{(A.13)}$$

When $\bar{\Delta} < 1$, the infinite sum converges and can be written in closed form:

$$S(\bar{\Delta}) \equiv \sum_{n=0}^{\infty} \frac{u_n^{\bar{\Delta}}}{2n + \Delta} = \frac{{}_3F_2(\bar{\Delta}, \bar{\Delta}, \Delta/2; 1, 1 + \Delta/2; 1)}{\Delta}, \quad \bar{\Delta} < 1. \quad \text{(A.14)}$$

*A.2.1  $K_{eff,2}$ for specific models*

For the Ising, we include only $\mathbb{1}$, i.e {div.} = {$\mathbb{1}$}, {conv.} = $\emptyset$ in (A.12), yeilding:

$$\langle f|i\rangle \frac{g^2 R^{3-2\Delta}}{2}[P(\Delta'_{T,i},\Delta)+P(\Delta'_{T,f},\Delta)]. \tag{A.15}$$

This indeed agrees with the result given in Eq. (16) with $g \to m$ and $\Delta = \Delta_\epsilon = 1$.

In the Tricritical to Critical Ising flow, we consider {div.} = {$\mathbb{1}$} and {conv.} = {$\epsilon'$} in (A.12), yeilding:

$$\begin{aligned}\left(K^{\text{Tric.}}_{\text{eff},2}\right)_{fi} = \langle f|i\rangle \frac{g^2 R^{3-2\Delta}}{2}\Big[P(\Delta'_{T,i},\Delta)+P(\Delta'_{T,f},\Delta)\Big] \\ + \langle f|\epsilon'(1)|i\rangle\, C^{\epsilon'}_{\Delta\Delta}\frac{g^2 R^{3-2\Delta}}{2}\Big[2S(\Delta/2)-P(\Delta'_{T,i},\Delta/2)-P(\Delta'_{T,f},\Delta/2)\Big].\end{aligned} \tag{A.16}$$

Once again, this agrees with Eq. (23) for $\Delta = \Delta_{\epsilon'} = 6/5$.

Finally, for the $M(3,7)$ flow, we include {div.} = {$\mathbb{1}, \phi_{1,3}$} and {conv.} = $\emptyset$, since there are no UV finite contributions from primaries to $K_{\text{eff},2}$ at order $\mathcal{O}(g_5^2)$. This time (A.12) reduces to

$$\begin{aligned}\left(K^{M(3,7)}_{\text{eff, }2}\right)_{fi} = \langle f|i\rangle \frac{g^2 R^{3-2\Delta}}{2}\Big[P(\Delta'_{T,i},\Delta)+P(\Delta'_{T,f},\Delta)\Big] \\ + \langle f|\phi_{1,3}(1)|i\rangle\, C^3_{\Delta\Delta}\frac{g^2 R^{3-2\Delta}}{2}\Big[P(\Delta'_{T,i},\Delta-\Delta_{1,3}/2)+P(\Delta'_{T,f},\Delta-\Delta_{1,3}/2)\Big].\end{aligned} \tag{A.17}$$

Once again this agrees with Eq. (36) with $g = g_5, \Delta = \Delta_{1,5} = 8/7$ and $\Delta_{1,3} = -2/7$.

## A.3  $K_{\text{eff},3}$

In this section we provide a derivation of the $K_{\text{eff},3}$ operators used in the Tricritical to Critical Ising flow as well as the $M(3,7)$ flow. At third order in the coupling, there are two distinct contributions to $H_{\text{eff},3}(\epsilon)$ computed for the $\epsilon$-regulated theory of (A.1):

$$(H_{\text{eff},3}(\epsilon))_{fi} = \big[(\Delta H_2(\epsilon))_{fi}+(\Delta H_3(\epsilon))_{fi}\big]_{\mathcal{O}(g^3)},$$

with the third term in the Effective Hamiltonian expansion reads:

$$(\Delta H_3(\epsilon))_{ij} = \frac{R^2}{2}\left[\frac{V(\epsilon)_{ih}V(\epsilon)_{hh'}V(\epsilon)_{h'j}}{\Delta_{ih}\Delta_{ih'}}-\frac{V(\epsilon)_{il}V(\epsilon)_{lh}V(\epsilon)_{hj}}{\Delta_{hi}\Delta_{hl}}+f\leftrightarrow i\right]. \tag{A.18}$$

In the above a sum over scaling dimensions $\Delta_h, \Delta_{h'} > \Delta_T$ and $\Delta_l \leqslant \Delta_T$ is implied. As we did for the second order result, we can write (A.18) as a time ordered integral over matrix elements on the cylinder:

$$\begin{aligned}(\Delta H_3(\epsilon))_{fi}\big|_{\mathcal{O}(g^3)} = \frac{g^3}{2(2\pi)^2}\int\limits_{\substack{T,\epsilon \\ 0<x_{1,2}\leqslant 2\pi R}} d\tau_1 d\tau_2 dx_1 dx_2 \\ \Bigg[\sum_{\Delta_h,\Delta_{h'}>\Delta_T} e^{\tau_2\frac{\Delta_{h'}-\Delta_h}{R}}e^{\tau_1\frac{\Delta_h-\Delta_i}{R}}\langle f|\phi_{0,\kappa}|h'\rangle\langle h'|\phi_{0,x_2}|h\rangle\langle h|\phi_{0,x_1}|i\rangle \\ -\sum_{\Delta_l\leqslant\Delta_T,\Delta_h>\Delta_T} e^{\tau_2\frac{\Delta_l-\Delta_i}{R}}e^{\tau_1\frac{\Delta_h-\Delta_l}{R}}\langle f|\phi_{0,\kappa}|h\rangle\langle h|\phi_{0,x_1}|l\rangle\langle l|\phi_{0,x_2}|i\rangle\Bigg],\end{aligned} \tag{A.19}$$

where $T$ indicates the time ordered domain of integration $-\infty < \tau_1 \leqslant \tau_2 \leqslant 0$, and we again use the $\epsilon$ subscript to schematically indicate that all correlators are to be regulated with $\epsilon$-balls. We use the abbreviation $\phi_{0,x} \equiv \phi_\Delta(0,x)$, and $\kappa \in [0, 2\pi R]$ is arbitrary. Using $\phi_\Delta(\tau,x) = e^{tH_0}\phi_\Delta(0,x)e^{-tH_0}$ and Weyl transforming the integral from the cylinder to the plane, we obtain

$$(\Delta H_3(\epsilon))_{fi}\big|_{\mathcal{O}(g^3)} = \frac{R^{5-3\Delta}g^3}{2(2\pi)^2} \int\limits_{\substack{\text{r.o},|1-x_{1,2}|>\epsilon \\ |x_1-x_2|>\epsilon|x_2|}} d^2x_1 d^2x_2 |x_1|^{\Delta-2}|x_2|^{\Delta-2}$$

$$\left[ \langle f|\phi_\Delta(1)\vdots\phi_\Delta(x_2)\vdots\phi_\Delta(x_1)|i\rangle - \sum_{\Delta_l \leqslant \Delta_T} \langle f|\phi_\Delta(1)\vdots\phi_\Delta(x_1)|l\rangle \langle l|\phi_\Delta(x_2)|i\rangle + f \leftrightarrow i \right],$$

(A.20)

where r.o refers to the radially ordered region $0 < |x_1| \leqslant |x_2| \leqslant 1$. In order to obtain $H_{\text{eff},3}(\epsilon)$, one has to add the $\mathcal{O}(g^3)$ contributions coming from $\Delta H_2(\epsilon)$. One can easily check that such contributions vanish when the only renormalized operator at second order in $g$ is the identity. Therefore we first proceed from (A.20) to derive the $K_{\text{eff},3}$ operator for the Tricritical to Critical Ising flow, and later revisit the computation in the case of the $M(3,7)$ flow. For both derivations, we will use the following generalized OPE:

$$\langle f|\phi_\Delta(1)\phi_\Delta(x_2)\phi_\Delta(x_1)|i\rangle = \sum_{\mathcal{O}} \langle f|\mathcal{O}(1)|i\rangle \langle \mathcal{O}(\infty)\phi_\Delta(1)\phi_\Delta(x_2)\phi_\Delta(x_1)\rangle, \quad \text{(A.21)}$$

valid in the region $|x_1| \leqslant |x_2| < 1, |1-x_1| < 1, |1-x_2| < 1$.

### A.3.1  $K_{\text{eff},3}$ for the Tricritical to Critical Ising flow

In the Tricritical to Critical Ising flow, we include only the $\mathcal{O} = \mathbb{1}$ contribution in (A.21). We will again make the approximation of replacing the correlators in (A.20) by (A.21) over the entire range of integration. In our case, since there are no UV divergences at third order, the integral of $\langle\phi_\Delta(1)\phi_\Delta(x_2)\phi_\Delta(x_1)\rangle$ without the partial resolutions of the identity is finite, and equals:[19]

$$\int_{\text{r.o}} d^2x_1 d^2x_2 \frac{|x_1|^{\Delta-2}|x_2|^{\Delta-2}C_{\Delta\Delta}^\Delta}{|1-x_1|^\Delta|1-x_2|^\Delta|x_1-x_2|^\Delta} = (2\pi)^2 C_{\Delta\Delta}^\Delta \sum_{Q,K=0}^\infty \frac{(A_{K,Q})^2}{(2K+\Delta)(2Q+\Delta)}, \quad \text{(A.22)}$$

where $A_{K,Q} = \sum_{k=0}^{\text{Min}(K,Q)} r_k^{\Delta/2} r_{K-k}^{\Delta/2} r_{Q-k}^{\Delta/2}$, $r_n^\alpha \equiv \Gamma(n+\alpha)/(\Gamma(n+1)\Gamma(\alpha))$. Comparing expressions (A.19) and (A.20) one can compute the partial insertions of the identity. The term with the two $\vdots$ insertions is computed by expanding the correlator in powers of $|x_2|$ and $|x_1|/|x_2|$, and keeping only powers of both variables higher than $\Delta_{T,i}$. This is equivalent to the restriction $Q, K > \Delta_{T,i}$ of the sum in (A.22). To compute the second term in the second line of (A.20), one expands the correlator in powers of $|x_1|$ and $|x_2|/|x_1|$.[20] The partial insertions amount to keeping one index smaller than $\Delta_{T,i}$ in the expansion. The final answer we obtain is (we

---

[19]This integral is UV finite under our assumptions so we can compute it setting $\epsilon = 0$.

[20]The integral of this expansion does not converge since $|x_2|/|x_1|$ is unbounded within the region of integration, however the partial insertions render the integral finite.

directly take the $\epsilon \to 0$ limit to obtain the $K_{\text{eff},3}$ matrix element):

$$
(K_{\text{eff},3})_{fi} = \frac{R^{5-3\Delta} g^3}{2} \langle f|i \rangle C_{\Delta\Delta}^{\Delta} \Bigg[ \sum_{Q,K > \Delta'_{T,i}/2} \frac{(A_{K,Q})^2}{(2K+\Delta)(2Q+\Delta)}
$$
$$
- \sum_{\substack{2K=0 \\ 2Q > \Delta'_{T,i}}}^{2K \leqslant \Delta'_{T,i}} \frac{(A_{K,Q})^2}{(2Q+\Delta)(2Q-2K)} + f \leftrightarrow i \Bigg]. \tag{A.23}
$$

This agrees with Eq. (25) upon substituting $\Delta = \Delta_{\epsilon'} = 6/5$, and noting that $A_{K,Q}$ has the following closed form:

$$
A_{K,Q} = \frac{\Gamma\left(\frac{1}{2}(\Delta+2K)\right)\Gamma\left(\frac{1}{2}(\Delta+Q)\right)\,{}_3F_2(\Delta/2,-Q,-K;1-\Delta/2-Q,1-\Delta/2-K;1)}{\Gamma^2(\Delta/2)\Gamma(1+K)\Gamma(1+Q)}. \tag{A.24}
$$

The $K_{\text{eff},3}$ matrix elements for this flow were computed numerically via integration in order to approximate certain infinite sums in (A.23) which have no useful closed form expression.

### A.3.2 $K_{eff,3}$ for the $M(3,7)$ flow

We now derive the $K_{\text{eff},3}$ operator for the $M(3,7)$ flow, and start by computing $H_{\text{eff},3}(\epsilon)$. Since $C_{55}^5 = 0$, there isn't any contribution from $\mathcal{O} = \mathbb{1}$ in (A.21), and the leading contribution is $\mathcal{O} = \phi_{1,3}$. The four-point function that we need, to compute this contribution is given by

$$
\langle \phi_{1,3}(\infty)\phi_{1,5}(1)\phi_{1,5}(z_2)\phi_{1,5}(z_1) \rangle = C_{335} C_{355} \frac{\left|1+z_1^2+z_2^2-z_1-z_2-z_1 z_2\right|^2}{|1-z_1|^{18/7}|1-z_2|^{18/7}|z_1-z_2|^{18/7}}, \tag{A.25}
$$

where we have used complex coordinates $z_i = x_i^0 + ix_i^1$. This minimal model four point function can be derived using the Coulomb gas method introduced in Refs. [34,35].

As before, in order to compute the first and second term on the second line of (A.20), we need to expand (A.25) in powers of $|z_2|, |z_1|/|z_2|$ and $|z_1|, |z_2|/|z_1|$ respectively. The series expansion of the four point function in $|z_2|, |z_1|/|z_2|$ reads

$$
C_{35}^3 C_{55}^3 \sum_{K,Q=0}^{\infty} \left(\frac{z_1}{z_2}\right)^K z_2^{Q-9/7} B_{K,Q} \times c.c, \tag{A.26}
$$

where $\times c.c$ refers to multiplication by the complex conjugate and we have defined $B_{K,Q} \equiv A_{K,Q} - A_{K-1,Q-1} - A_{K,Q-1} + A_{K-2,Q-2} - A_{K-1,Q-2} + A_{K,Q-2}$ with

$$
A_{K,Q} = \sum_{k=0}^{\min(K,Q)} r_k^{9/7} r_{K-k}^{9/7} r_{Q-k}^{9/7} = \frac{\Gamma(K+\frac{9}{7})\Gamma(Q+\frac{9}{7})\,{}_3F_2(\frac{9}{7},-K,-Q;-K-\frac{2}{7},-Q-\frac{2}{7};1)}{\Gamma(\frac{9}{7})^2\Gamma(K+1)\Gamma(Q+1)}. \tag{A.27}
$$

Comparing with the same expression on the cylinder and matching powers of $e^{\tau_2}, e^{\tau_1-\tau_2}$ with those of $|z_2|, |z_1|/|z_2|$ we obtain

$$
\int_{\text{r.o.}} \left(\prod_{i=1}^{2} d^2 z_i |z_i|^{8/7-2}\right) \langle \phi_{1,3}(\infty)\phi_{1,5}(1)\,\vdots\,\phi_{1,5}(z_2)\,\vdots\,\phi_{1,5}(z_1) \rangle =
$$
$$
(2\pi)^2 C_{35}^3 C_{55}^3 \sum_{\substack{2K > \Delta'_{T,i} \\ 2Q > \Delta''_{T,i}}} \frac{(B_{K,Q})^2}{(2K+8/7)(2Q-2/7)}, \tag{A.28}
$$

where we define $\Delta'_{T,i} \equiv \Delta_T - \Delta_i - 8/7$ and $\Delta''_{T,i} \equiv \Delta_T - \Delta_i + 2/7$. A similar analysis yields, for the second term on the second line of (A.20):

$$\int_{\text{r.o.}} \left( \prod_{i=1}^{2} d^2 z_i |z_i|^{8/7-2} \right) \langle \phi_{1,3}(\infty) \phi_{1,5}(1) \mathop{\vdots}_i \phi_{1,5}(z_1) \mathop{\vdots}_i \phi_{1,5}(z_2) \rangle = \\ (2\pi)^2 C_{35}^3 C_{55}^3 \sum_{\substack{2Q > \Delta''_{T,i} \\ K=0}}^{2K \leqslant \Delta'_{T,i}} \frac{(B_{K,Q})^2}{(2Q - 2K - 10/7)(2Q - 2/7)}, \tag{A.29}$$

where we have denoted $\mathop{\vdots}_i \equiv \sum_{\Delta_l \leqslant \Delta'_{T,i}} |l\rangle \langle l|$.

We now compute the second contribution to $H_{\text{eff},3}(\epsilon)$, i.e. $\Delta H_2(\epsilon)\big|_{\mathcal{O}(g^3)}$. After Weyl transformation to the plane, the contribution from the local operator $\phi_{1,3}$ reads:

$$(\Delta H_2(\epsilon))_{fi}\big|_{\mathcal{O}(g^3)} = -\frac{g^3}{(2\pi)^2} \langle f | \phi_{1,3}(1) | i \rangle \lambda_{\text{c.t.}}^{\phi_{1,3}}(\epsilon) \int_{\substack{|x| \leqslant 1 \\ |1-x| > \epsilon}} d^2 x |x|^{\Delta - 2} \Bigg[ \langle \phi_{1,3}(\infty) \phi_{1,3}(1) \mathop{\vdots}_i \phi_\Delta(x) \rangle \\ + |x|^{\Delta_{1,3} - \Delta - 2} \langle \phi_{1,3}(\infty) \phi_\Delta(1) \mathop{\vdots}_i \phi_{1,3}(x) \rangle \Bigg]. \tag{A.30}$$

As we have seen in Subsec. A.1 and A.2 , both the counterterm coupling and the integral with the partial resolution of the identity can be written as a sum over $u_n$'s, and we obtain

$$(\Delta H_2(\epsilon))_{fi}\big|_{\mathcal{O}(g^3)} = -\frac{g^3 R^{5-3\Delta}}{2} \langle f | \phi_{1,3}(1) | i \rangle C_{35}^3 C_{55}^3 \sum_{n=0}^{\infty} \frac{u_n^{9/7,\epsilon}}{2n + 8/7} \\ \left( \sum_{2n > \Delta'_{T,i}} \frac{u_n^{4/7,\epsilon}}{2n + 8/7} + \sum_{2n > \Delta''_{T,i}} \frac{u_n^{4/7,\epsilon}}{2n - 2/7} + i \leftrightarrow f \right). \tag{A.31}$$

The final answer is

$$(K_{\text{eff},3})_{fi} = \frac{g^3 R^{5-3\Delta}}{2} \langle f | \phi_{1,3}(1) | i \rangle C_{35}^3 C_{55}^3 \Bigg[ \sum_{\substack{2K > \Delta'_{T,i} \\ 2Q > \Delta''_{T,i}}} \frac{(B_{K,Q})^2 - u_K^{4/7} u_Q^{9/7} - u_Q^{4/7} u_K^{9/7}}{(2K + 8/7)(2Q - 2/7)} \\ - \sum_{\substack{2Q > \Delta''_{T,i} \\ K=0}}^{2K \leqslant \Delta'_{T,i}} \frac{(B_{K,Q})^2}{(2Q - 2K - 10/7)(2Q - 2/7)} - \sum_{\substack{2K > \Delta'_{T,i} \\ Q=0}}^{2Q \leqslant \Delta''_{T,i}} \frac{u_K^{4/7} u_Q^{9/7}}{(2K + 8/7)(2Q + 8/7)} \\ - \sum_{\substack{2Q > \Delta''_{T,i} \\ K=0}}^{2K \leqslant \Delta'_{T,i}} \left[ \frac{(B_{K,Q})^2}{(2Q - 2K - 10/7)(2Q - 2/7)} + \frac{u_K^{9/7} u_Q^{4/7}}{(2K + 8/7)(2Q - 2/7)} \right] \\ + \frac{10}{7} \sum_{\substack{2K > \Delta'_{T,i} \\ 2Q > \Delta''_{T,i}}} \frac{u_k^{4/7} u_Q^{9/7}}{(2K + 8/7)(2Q - 2/7)(2Q + 8/7)} + i \leftrightarrow f \Bigg]. \tag{A.32}$$

This coincides with Eq. (38) upon setting $g = g_5$ and $\Delta = \Delta_{1,5} = 8/7$.[21] Since not all infinite sums appearing in (A.32) have a closed form expression suited to efficient numerical evaluation, we computed $K_{\text{eff},3}$ by performing numerical extrapolations of infinite sums.

## B   Energy Levels in the Ising$+\epsilon$ QFT

In this appendix, we provide results for the energy levels of the massive Majorana fermion QFT in two dimensions. Some are equivalent to those in the Ising CFT deformed with its $\epsilon$ operator, so the formulae shown here may be directly compared with the HT results presented in Sec. 4.2.

The ground state energy for the 2d fermion on the spatial circle with radius $R$ is related to $f(T)$, the free energy per unit length of the 2d fermion at temperature $T = 1/(2\pi R)$ in infinite spatial volume

$$E_0(R) = 2\pi R f(R). \tag{B.1}$$

This can be seen in Euclidean signature by interchanging the space and time directions – see for example [17]. Since the free energy may be expressed using the logarithm of the partition function, the ground state energy of the free fermion takes the form [36]

$$E_0(R) = E_{\text{bulk}}(R) - |m| \int_{-\infty}^{\infty} \frac{d\theta}{2\pi} \cosh\theta \log\left(1 + e^{-2\pi|m|R\cosh\theta}\right). \tag{B.2}$$

In this expression, $E_{\text{bulk}}(R)$ is the contribution from zero-point energies. Naively it is a UV divergent quantity. We must regularize and calculate it using our choice of renormalization scheme. It takes the form

$$E_{\text{bulk}}(R) = -R \int dp \left[\sqrt{p^2 + m^2} - p\right] + V_{\text{c.t}}^{\mathbb{1}}, \tag{B.3}$$

where $V_{\text{c.t}}^{\mathbb{1}}$ denotes the contribution from the identity operator counterterm, which is

$$V_{\text{c.t}}^{\mathbb{1}} = m^2 R \sum_{n=0}^{} \frac{1}{2n+1}. \tag{B.4}$$

The $p$ term in the square parentheses of (B.3) ensures that $E_{\text{bulk}}$ vanishes when $m = 0$, as it does in the free Ising CFT.

The terms in (B.3) require regularization. It is most straightforward to use a local momentum cutoff regulator, so that $p \leqslant N_{\max}/R$. Each term in the sum of (B.4) can be interpreted as coming from a loop of massless fermions with momentum $(n + 1/2)/R$, so the sum should be regulated using $N_{\max}$ also. Therefore we have

$$E_{\text{bulk}}(R) = \lim_{N_{\max} \to \infty} \left\{ -\frac{1}{R} \int_0^{N_{\max}} dn \left[\sqrt{n^2 + m^2 R^2} - n\right] + m^2 R \sum_{n=0}^{N_{\max}} \frac{1}{2n+1} \right\}. \tag{B.5}$$

After taking the limit, we find that for our choice of renormalization scheme, the ground state energy is given by

$$E_0(R) = -\frac{m^2 R}{4} \left(1 - \log(2m^2 R^2) - 2\gamma\right) - |m| \int_{-\infty}^{\infty} \frac{d\theta}{2\pi} \cosh\theta \log\left(1 + e^{-2\pi|m|R\cosh\theta}\right). \tag{B.6}$$

---

[21]Note that Eq. (38) was written in a more compact form in which individual infinite sums diverged, but the entire expression was finite. In Eq. (A.32), terms have been combined in a way that each sum is convergent.

where $\gamma$ is the Euler-Mascheroni constant.

The excited states each belong to one of two subsectors - the Neveu Schwartz (NS) and Ramond (R) sectors. The two subsectors are distinguished by the boundary conditions satisfied by the fermion fields on the circle; in the NS sector, fermion fields are periodic and single particle excitations have quantized momenta with $p_n = (n + 1/2)/R$ for $n \in \mathbb{Z}$, whereas in the R sector particles have momenta $k_n = n/R$.

In the NS sector, the excited state energies take the form

$$E(R) = E_0(R) + \sum_{i=1}^{N} \sqrt{p_{n_i}^2 + m^2}\,. \tag{B.7}$$

The number of particle excitations $N$ must be even for $m > 0$ [36].

In the R sector, the excited state energies take the form

$$E(R) = E_0(R) + \Delta + \sum_{i=1}^{N} \sqrt{k_{n_i}^2 + m^2}\,. \tag{B.8}$$

In this case, the number of excitations $N$ must be odd. The quantity $\Delta$ represents the splitting between the R sector vacuum state and the true ground state $E_0(R)$, which belongs to the NS sector. $\Delta$ is an exponentially suppressed function of $mR$.

In Fig. 1, we calculate the gaps between excited state and the ground state energies for $mR = 2.0$. For this value, $\Delta$ is negligible. The first four excited states are as follows: The lowest is in the R sector corresponds to a single fermion with zero momentum. For largish $mR$, it has a gap given by $E_1 - E_0 \approx m$. The second and third excited states belong to the NS sector. They are two fermion states with equal (and opposite) momenta of magnitude $p = 1/(2R)$ and $p = 3/(2R)$. Their gaps are given by $E_2 - E_0 = 2m\sqrt{1 + 1/(2mR)^2}$ and $E_3 - E_0 = 2m\sqrt{1 + 9/(2mR)^2}$ respectively. The fourth excited state is in the R sector and contains one fermion with zero momentum and two fermions with equal and opposite momenta of magnitude $k = 1/R$. It has a gap of $E_4 - E_0 \approx m(1 + 2\sqrt{1 + 1/(mR)^2})$.

## C Fusion rules of selected minimal models

In this appendix we give the fusion rules for minimal models of interest in this work, namely the $M(4,5)$ (Tricritical Ising model), $M(3,7)$ and $M(3,5)$.

### C.1 Tricritical Ising Model

The fusion rules for the Tricritical Ising model read:

$$\phi_{1,2} \times \phi_{1,2} \sim \mathbb{1} + \phi_{1,3} \qquad \phi_{1,2} \times \phi_{1,3} \sim \phi_{1,2} + \phi_{1,4}$$
$$\phi_{1,2} \times \phi_{1,4} \sim \phi_{1,3} \qquad \phi_{1,2} \times \phi_{2,2} \sim \phi_{2,2} + \phi_{2,1}$$
$$\phi_{1,2} \times \phi_{2,1} \sim \phi_{2,2} \qquad \phi_{1,3} \times \phi_{1,3} \sim \mathbb{1} + \phi_{1,3}$$
$$\phi_{1,3} \times \phi_{1,4} \sim \phi_{1,2} \qquad \phi_{1,3} \times \phi_{2,2} \sim \phi_{2,2} + \phi_{2,1}$$
$$\phi_{1,3} \times \phi_{2,1} \sim \phi_{2,2} \qquad \phi_{1,4} \times \phi_{1,4} \sim \mathbb{1}$$
$$\phi_{1,4} \times \phi_{2,2} \sim \phi_{2,2} \qquad \phi_{1,4} \times \phi_{2,1} \sim \phi_{2,1}$$
$$\phi_{2,2} \times \phi_{2,2} \sim \mathbb{1} + \phi_{1,2} + \phi_{1,3} + \phi_{1,4} \qquad \phi_{2,2} \times \phi_{2,1} \sim \phi_{1,2} + \phi_{1,3}$$
$$\phi_{2,1} \times \phi_{2,1} \sim \mathbb{1} + \phi_{1,4}.$$

The non trivial OPE coefficients are

$$C^{(1,3)}_{(1,2),(1,2)} = \frac{2}{3}\sqrt{\frac{\Gamma(4/5)\Gamma(2/5)^3}{\Gamma(1/5)\Gamma(3/5)^3}} \qquad C^{(1,3)}_{(1,3),(1,3)} = C^{(1,3)}_{(1,2),(1,2)}$$

$$C^{(1,3)}_{(1,2),(1,4)} = 3/7 \qquad C^{(1,2)}_{(2,1),(2,2)} = 1/2$$

$$C^{(1,2)}_{(2,2),(2,2)} = 3/2\, C^{(1,3)}_{(1,2),(1,2)} \qquad C^{(1,3)}_{(2,1),(2,2)} = 3/4$$

$$C^{(1,3)}_{(2,2),(2,2)} = 1/4\, C^{(1,3)}_{(1,2),(1,2)} \qquad C^{(1,4)}_{(2,2),(2,2)} = 1/56$$

$$C^{(1,4)}_{(2,1),(2,1)} = 7/8.$$

### C.2  $M(3,7)$

The fusion rules of the $M(3,7)$ model are:

$$\phi_{1,2} \times \phi_{1,2} \sim \mathbb{1} + i\phi_{1,3} \qquad \phi_{1,2} \times \phi_{1,3} \sim i\phi_{1,2} + \phi_{1,4}$$
$$\phi_{1,2} \times \phi_{1,4} \sim \phi_{1,3} + i\phi_{1,5} \qquad \phi_{1,2} \times \phi_{1,5} \sim i\phi_{1,4} + \phi_{1,6}$$
$$\phi_{1,2} \times \phi_{1,6} \sim \phi_{1,5} \qquad \phi_{1,3} \times \phi_{1,3} \sim \mathbb{1} + i\phi_{1,3} + \phi_{1,5}$$
$$\phi_{1,3} \times \phi_{1,4} \sim \phi_{1,2} + i\phi_{1,4} + \phi_{1,6} \qquad \phi_{1,3} \times \phi_{1,5} \sim \phi_{1,3} + i\phi_{1,5}$$
$$\phi_{1,3} \times \phi_{1,6} \sim \phi_{1,4} \qquad \phi_{1,4} \times \phi_{1,4} \sim \mathbb{1} + i\phi_{1,3} + \phi_{1,5}$$
$$\phi_{1,4} \times \phi_{1,5} \sim i\phi_{1,2} + \phi_{1,4} \qquad \phi_{1,4} \times \phi_{1,6} \sim \phi_{1,3}$$
$$\phi_{1,5} \times \phi_{1,5} \sim \mathbb{1} + i\phi_{1,3} \qquad \phi_{1,5} \times \phi_{1,6} \sim \phi_{1,2}$$
$$\phi_{1,6} \times \phi_{1,6} \sim \mathbb{1}.$$

We have indicated with a factor of $i$ the OPE coefficients which are purely imaginary, to distinguish them from the real ones. In (C.1), we give the value of the non trivial OPE coefficients between primaries in the even sector:

$$C_{333} = i\sqrt{-\frac{\Gamma(1/7)^3\Gamma(3/7)^3\Gamma(5/7)^3\Gamma(8/7)}{\Gamma(-1/7)\Gamma(2/7)^3\Gamma(4/7)^3\Gamma(6/7)^3}} \qquad \approx \quad 1.97409i,$$

$$C_{355} = i\sqrt{-\frac{\Gamma(-5/7)^2\Gamma(1/7)\Gamma(3/7)^3\Gamma(5/7)\Gamma(8/7)\Gamma(11/7)^2}{\Gamma(-4/7)^2\Gamma(-1/7)\Gamma(2/7)\Gamma(4/7)^3\Gamma(6/7)\Gamma(12/7)^2}} \qquad \approx \quad 0.979627i, \qquad \text{(C.1)}$$

$$C_{335} = -\sqrt{\frac{\Gamma(-5/7)\Gamma(-2/7)\Gamma(3/7)\Gamma(5/7)^2\Gamma(6/7)\Gamma(8/7)^2}{\Gamma(-1/7)^2\Gamma(1/7)\Gamma(2/7)^2\Gamma(4/7)\Gamma(9/7)\Gamma(12/7)}} \qquad \approx \quad -0.113565.$$

These constants (including one nontrivial relative sign between $C_{333}$ and $C_{355}$) can be determined from the four-point functions $\langle \phi_{1,3} \phi_{1,3} \phi_{1,3} \phi_{1,3} \rangle$ and $\langle \phi_{1,3} \phi_{1,3} \phi_{1,3} \phi_{1,5} \rangle$, which we computed using the Coulomb gas method [34, 35].

### C.3 $M(3,5)$

Here we give the fusion rules of the $M(3,5)$ model.

$$\phi_{1,2} \times \phi_{1,2} \sim \mathbb{1} + i\phi_{1,3} \qquad\qquad \phi_{1,2} \times \phi_{1,3} \sim i\phi_{1,2} + \phi_{1,4} \qquad\qquad \text{(C.2)}$$

$$\phi_{1,2} \times \phi_{1,4} \sim \phi_{1,3} \qquad\qquad\qquad \phi_{1,3} \times \phi_{1,3} \sim \mathbb{1} \qquad\qquad\qquad\;\; \text{(C.3)}$$

$$\phi_{1,3} \times \phi_{1,4} \sim \phi_{1,2} \qquad\qquad\qquad \phi_{1,4} \times \phi_{1,4} \sim \mathbb{1}. \qquad\qquad\quad\;\; \text{(C.4)}$$

## D Review of $\mathcal{PT}$ Symmetry

In this appendix we review in detail the $\mathcal{PT}$ symmetry present in the $M(3,7)$ flow and what impact it has on the spectrum.

An obvious consequence of imaginary OPE coefficients in the $M(3,7)$ CFT is that the QFT Hamiltonian (35) is not a Hermitian operator. Nevertheless such a non-Hermitian Hamiltonian can have a real spectrum bounded from below provided it is $\mathcal{PT}$-symmetric, see e.g. [37] for a comprehensive review. We now summarize the main facts about $\mathcal{PT}$-symmetry which are relevant for the analysis of the $M(3,7) + i\phi_{1,3} + \phi_{1,5}$ theory studied in the main text. The discrete symmetry is represented on the Hilbert space of the theory by an antiunitary involution operator ($(\mathcal{PT})^2 = \mathbb{1}$ and $\mathcal{PT}\lambda\mathcal{PT} = \lambda^*$ for any $\lambda \in \mathbb{C}$).

There are two conditions that define unbroken $\mathcal{PT}$ symmetry. The *first* is the usual definition of a symmetry of a quantum system, i.e.

$$[\mathcal{PT}, H] = 0. \qquad\qquad\qquad\qquad\qquad \text{(D.1)}$$

The *second* is that any eigenstate of the Hamiltonian $|\psi\rangle$ is also an eigenstate of $\mathcal{PT}$ [22] with eigenvalue one:[23]

$$\mathcal{PT}|\psi\rangle = |\psi\rangle, \qquad (H|\psi\rangle = E|\psi\rangle). \qquad\qquad \text{(D.2)}$$

Assuming (D.1) and (D.2) it is straightforward to show that any eigenvalue $E$ of the Hamiltonian must be real:

$$E\lambda|\psi\rangle = H\mathcal{PT}|\psi\rangle = \mathcal{PT}H|\psi\rangle = \mathcal{PT}E|\psi\rangle = \mathcal{PT}E\mathcal{PT}\mathcal{PT}|\psi\rangle = E^*\lambda|\psi\rangle.$$

We now argue that the $M(3,7)$ CFT has unbroken $\mathcal{PT}$ in the sense of (D.1) and (D.2). First, we note that the eigenvalues of the CFT Hamiltonian on the cylinder are real. Second,

---

[22]In the case of a linear symmetry operator, equations (D.1) and (D.2) are famously equivalent: any commuting (diagonalizable) linear operators can be simultaneously diagonalized. However this theorem does not hold for antilinear operators, hence the two distinct conditions.

[23]The eigenvalues of antiunitary involutions are pure phases $e^{i\alpha}, \alpha \in \mathbb{R}$, and if the Hamiltonian and the $\mathcal{PT}$ operator are simultaneously diagonalizable, one can always define Hamiltonian eigenstates such that their eigenvalue under $\mathcal{PT}$ is one. Indeed, given an eigenvector of the Hamiltonian $|\psi\rangle$ such that $\mathcal{PT}|\psi\rangle = e^{i\alpha}|\psi\rangle$, then $|\psi'\rangle \equiv e^{i\alpha/2}|\psi\rangle$ will have eigenvalue one:

$$\mathcal{PT}|\psi'\rangle = \mathcal{PT}e^{i\alpha/2}|\psi\rangle = e^{-i\alpha/2}\mathcal{PT}|\psi\rangle = e^{i\alpha/2}|\psi\rangle = |\psi'\rangle.$$

the CFT states are eigenstates of the $\mathcal{PT}$ operator defined in 6, with eigenvalues given in Tab. 3 (the descendants of a given conformal family have the same eigenvalues as their primary operator). Together these two facts imply (D.1), since:

$$\mathcal{PT}H_{\mathrm{CFT}}|\mathcal{O}_{\mathrm{CFT}}\rangle = \mathcal{PT}E_{\mathcal{O}}|\mathcal{O}_{\mathrm{CFT}}\rangle = E_{\mathcal{O}}(-1)^{\mathrm{PT}_{\mathcal{O}}}|\mathcal{O}_{\mathrm{CFT}}\rangle = H_{CFT}\mathcal{PT}|\mathcal{O}_{\mathrm{CFT}}\rangle,$$

Where $E_{\mathcal{O}} = (\Delta_{\mathcal{O}} - c/12)/R$. Since this is true for all CFT states, the commutator vanishes identically. [24]

Now we discuss the deformations to the UV CFT. The primary $\phi_{1,5}$ is even under $\mathcal{PT}$ while $\phi_{1,3}$ is odd, so adding to the CFT Hamiltonian $g_5\phi_{1,5} + ig_3\phi_{1,3}$ will maintain condition (D.1) along the RG flow (with $g_3, g_5 \in \mathbb{R}$). Therefore, a real spectrum along the RG flow is now equivalent to condition (D.2) being satisfied. It is clear that close to the UV fixed point, conformal perturbation theory will always predict real energy levels, so we expect (D.2) to hold for small coupling. However, at strong coupling (D.2) may no longer hold (see for example an illustrative two dimensional quantum mechanical toy model in [37]), in which case the spectrum will develop complex eigenvalues.

Complex eigenvalues come in conjugate pairs, and the corresponding Hamiltonian eigenstates are related under $\mathcal{PT}$: [25]

$$\mathcal{PT}|E\rangle = |E^*\rangle, \tag{D.3}$$

where $H|E\rangle = E|E\rangle, \quad H|E^*\rangle = E^*|E^*\rangle$. Equation (D.3) can be shown to hold with a very similar argument to the one used to show the reality of the spectrum under unbroken $\mathcal{PT}$ symmetry:

$$H\mathcal{PT}|E\rangle = \mathcal{PT}H|E\rangle = \mathcal{PT}E|E\rangle = E^*\mathcal{PT}|E\rangle \implies \mathcal{PT}|E\rangle \propto |E^*\rangle,$$

Where we have only used (D.1) and the properties of the $\mathcal{PT}$ operator.

In summary, the Hamiltonian of our QFT is $\mathcal{PT}$ invariant, which implies either a real spectrum or complex conjugate pairs of eigenvalues. For small values of the couplings, where conformal perturbation theory accurately predicts the spectrum, $\mathcal{PT}$ symmetry is unbroken, hence the spectrum is real and energy eigenstates can be defined to be invariant under $\mathcal{PT}$. Finally, in some regions of the two dimensional phase diagram spanned by the two couplings, we may expect $\mathcal{PT}$ symmetry to be broken. This entails pairs of eigenvalues becoming complex conjugates of each other, and in that case the corresponding eigenvectors get mapped to each other under $\mathcal{PT}$.

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
