# Peer review of "Hamiltonian Truncation Crafted for UV-divergent QFTs"

_SciPost Physics_

## Round 1 · Referee Report · Anonymous · 2024-2-13

Report
This is a good paper suitable for the journal. I recommend it for publication after the authors respond to some comments and suggestions which are in the attached report.
Requested changes
See the report attached.

---

## Round 1 · Referee Report · Anonymous · 2024-2-18

Strengths
1. The subject is relevant for ongoing research.
2. The goals are clearly stated, and the exposition is logical and easy to follow.
3. The results are interesting and expected to be useful for future studies.
4. The methodological developments are well-supported by the result of explicit calculations combining numerical and analytic methods.
Weaknesses
1. The paper is very technical, and its only new non-technical result is establishing a new (but fully expected) flow between minimal models M(3,7) and M(3,5).
2. It is mostly incremental work refining previous results in the field, mostly relevant for those working in numerical implementation of truncated conformal space approach in 2 dimensions.
3. Some nontrivial procedures are neither explained nor any references are given (see report).
Report
This paper reports a development of methodology for truncated conformal space approach. This is important for practitioners of the approach, but the downside is that the physics content is thin. As a methodology development, it is incremental rather than groundbreaking and therefore better fits Scipost Physics Core's acceptance criteria.
This work is very useful as a guide to improving the truncation dependence for UV divergent theories. The only omission is that the rather nontrivial algorithmic issues of removing Virasoro null states and computing the matrix elements are only mentioned in passing, leaving the reader without guidance for their implementation. This could have been solved by including a suitable reference: e.g., the procedure was recently been described by Horvath et al. in Computer Physics Communications 277: 108376, together with other important algorithmic developments which utilise chiral factorisation to improve computational efficiency.
Another similar issue is related to the effective field theory approach, which (to my knowledge) was first applied to this class of systems by Al. B. Zamolodchikov in his seminal work Nuclear Physics B358 (1991) 524-546. The paper itself is cited as Ref. 21, but the fact that it contains the effective field theory is relegated to a footnote, which does not give justice to its importance. From the TT-bar point of view, another recent relevant work (besides the already included Refs. 26 and 27) is C. Ahn and A. Leclair, JHEP 08 (2022) 179.
Requested changes
1. Improve the self-contained nature of the paper by helping the reader find some guide to the necessary CFT methodology.
2. Add references (see report).

---

## Round 1 · Referee Report · Anonymous · 2024-2-21

Strengths
i. The manuscript is clearly written.
ii. It studies interesting examples of Hamiltonian truncation.
iii. It is important to consider cases studied using Hamiltonian truncation where counterterms are needed to render the theory finite.
Weaknesses
I have queries of the authors but I wouldn't say my questions represent weaknesses.
Report
In this work the authors study Hamiltonian truncation in the context of 1+1d quantum field theories that arise from perturbing a conformal field theory (CFT) with a less relevant operator (an operator whose conformal scaling dimension is 1 or greater). Such theories typically have UV divergences that require counterterms to be added to Hamiltonians in order to render the theories finite.
The authors begin their treatment of such theories by introducing a regulator. Here the authors choose as a regulator the rule that spacetime points in a correlation function must be separated by a minimum distance $\epsilon$. This choice of regulator then for the cases considered leads to a finite spectrum in perturbation theory.
Having regulated the theory, the authors then proceed as it typical in Hamiltonian truncation. The authors introduce a cutoff $\Delta_T$ of the CFT computational basis -- this UV cutoff is different than the short distance regulator $\epsilon$. The authors then add terms to Hamiltonian meant to improve convergence in $\Delta_T$. What they notice is that once these terms are added, one can take the $\epsilon\rightarrow 0$ limit (i.e. remove the regulator) and be left with finite terms in the Hamiltonian. This leads to a simplification. Rather than having to take a double limit $\epsilon \rightarrow 0$ ${\bf and}$ $\Delta_T \rightarrow \infty$ in analyzing the numerical data, they only have to consider the later limit.
With this scheme so defined, the authors treat three examples: i) the Ising CFT perturbed by the thermal perturbation; ii) the tricritical Ising perturbed by the subleading energy operator; iii) the non-unitary CFT M(3,7) perturbed by Z2 even operators. In the last two cases, the perturbation takes the theory in the IR to the vicinity of another CFT. In the case of ii), the IR theory is near the c=1/2 Ising CFT. In the case of iii), it is near another non-unitary CFT M(3,5).
It is impressive that the authors can observe these non-trivial flows.
Overall I have a positive impression of this manuscript. I do have a few requested clarifications and modifications before recommending it for publication in SciPost.
Requested changes
1. The authors do not provide a clear rational to my eye for why they are introducing two UV cutoffs -- the typical cutoff that appears in Hamiltonian truncation, $\Delta_T$, but also a short distance cutoff, $\epsilon$. Given that the authors take $\epsilon$ to 0 before any numerics is performed, why introduce it all? Wouldn't a finite $\Delta_T$ be sufficient to permit the introduction of counterterms?
2. In studying the flow of the perturbed M(3,7) theory, it is unclear to me whether the expected flow is to M(3,5) or M(3,5) perturbed by $\phi_{1,3}$.
The authors say that for couplings of the perturbations that correspond to the angle $\alpha =\pi$, the flow is to an EFT that is given by M(3,5) perturbed by $\phi_{1,3}$. But it isn't clear to me whether I can choose $\alpha$ so that the flow is to pure M(3,5) (plus irrelevant terms). If I can choose $\alpha$ so that I'm flowing in the IR to a purely critical theory, I would want to know this $\alpha$ to greater precision than merely somewhere near $\alpha=\pi$.
3. A more minor comment: the labels/text in the figures are too small -- they could easily be made much larger for improved readability.

---

## Editorial Decision

unknown